# SeaFlux: harmonization of air-sea CO₂ fluxes from surface pCO₂ data products using a standardised approach

Amanda R. Fay[1*], Luke Gregor[2*], Peter Landschützer[3], Galen A. McKinley[1], Nicolas Gruber[2], Marion Gehlen[4], Yosuke Iida[5], Goulven G. Laruelle[6], Christian Rödenbeck[7], Alizée Roobaert[6], Jiye Zeng[8]

[1] Columbia University and Lamont Doherty Earth Observatory, Palisades NY, USA

[2] Institute of Biogeochemistry and Pollutant Dynamics, ETH Zurich, Zürich, Switzerland

[3] Max Planck Institute for Meteorology, 20146 Hamburg, Germany

[4] Laboratoire des Sciences du Climat et de l'Environnement, Institut Pierre Simon Laplace, Gif-Sur-Yvette, France

[5] Atmosphere and Ocean Department, Japan Meteorological Agency, 1-3-4 Otemachi, Chiyoda-Ku, Tokyo 100-8122, Japan

[6] Department of Geosciences, Environment & Society-BGEOSYS, Université Libre de Bruxelles, Brussels, CP160/02, Belgium

[7] Biogeochemical Signals, Max Planck Institute for Biogeochemistry, P.O. Box 600164, Hans-Knöll-Str. 10, 07745 Jena, Germany

[8] National Institute for Environmental Studies (NIES), 16-2 Onogawa, Tsukuba, Ibaraki, 305-8506, Japan

*ARF and LG contributed equally to this work as first authors.

*Correspondence to*: Amanda R. Fay (afay@ldeo.columbia.edu) and/or Luke Gregor (luke.gregor@usys.ethz.ch)

**Abstract.** Air-sea flux of carbon dioxide ($CO_2$) is a critical component of the global carbon cycle and the climate system with the ocean removing about a quarter of the $CO_2$ emitted into the atmosphere by human activities over the last decade. A common approach to estimate this net flux of $CO_2$ across the air-sea interface is the use of surface ocean $CO_2$ observations and the computation of the flux through a bulk parameterization approach. Yet, the details for how this is done in order to arrive at a global ocean $CO_2$ uptake estimate vary greatly, unnecessarily enhancing the spread. This resource enables users to harmonize an ensemble of products that interpolate surface ocean $CO_2$ observations to near-global coverage with a common methodology to fill in missing areas in the products. Further, the dataset provides the inputs to calculate fluxes in a consistent way with which we present an ensemble product. The ensemble data product, SeaFlux (Gregor & Fay, 2021), https://doi.org/10.5281/zenodo.5148795, https://github.com/luke-gregor/pySeaFlux), accounts for the diversity of the underlying mapping methodologies. Utilizing six global observation-based mapping products (CMEMS-FFNN, CSIR-ML6, JENA-MLS, JMA-MLR, MPI-SOMFFN, NIES-FNN), the SeaFlux ensemble approach adjusts for methodological inconsistencies in flux calculations. We address differences in spatial coverage of the surface ocean $CO_2$ between the mapping products which ultimately yields an increase in $CO_2$ uptake of up to 17% for some products. Fluxes are calculated using three wind products (CCMPv2, ERA5, and JRA55). Application of a scaled gas exchange coefficient has a greater impact on the resulting flux than solely the choice of wind product. With these adjustments, we present an ensemble of global surface ocean $pCO_2$ and air-sea carbon flux estimates. This work aims to support the community effort to perform model-data intercomparisons which will help to identify missing fluxes as we strive to close the global carbon budget.

## 1 Introduction

Surface ocean partial pressure of $CO_2$ ($pCO_2$) observations play a key role in constraining the global ocean carbon sink. This is because variation in surface ocean $pCO_2$, ultimately driven by increases in atmospheric $pCO_2$ levels, is the driving force governing the exchange of $CO_2$ across the air-sea interface, which is commonly described through a bulk formula (Garbe et al. 2014; Wanninkhof 2014):

$$Flux = k_w \cdot sol \cdot (pCO_2 - pCO_2^{atm}) \cdot (1 - ice) \tag{1}$$

where $k_w$ is the gas transfer velocity, *sol* is the solubility of $CO_2$ in seawater, in units mol m$^{-3}$ µatm$^{-1}$, $pCO_2$ is the partial pressure of surface ocean $CO_2$ in µatm, and $pCO_2^{atm}$ in units of µatm represents the partial pressure of atmospheric $CO_2$ in the marine boundary layer. Finally, to account for the seasonal ice cover in high latitudes, the fluxes are weighted by 1 minus the ice fraction (*ice*), *i.e.* the open ocean fraction.

With the increasing number of observations of $pCO_2$ available in each new release of the Surface Ocean Carbon Dioxide Atlas (SOCAT; Bakker et al. 2016) and the adoption of various $pCO_2$ mapping techniques, multiple observation-based estimates of the $pCO_2$ field are now publicly available and updated on an annual basis. Despite these advancements, the intercomparison of the products' global and regional flux values is hindered (1) by different areal coverage and (2) by a lack of a consistent approach to calculate the sea-air $CO_2$ flux from $pCO_2$ (Table A1). These differences in flux calculations, specifically differing spatial coverage, complicate comparisons between the products and Global Ocean Biogeochemistry Models (GOBM). In this work, we harmonize these products' flux estimates, specifically addressing three key differences between product methodologies. The resulting flux estimates can then be more directly compared with respect to uncertainty attribution with no source of difference that is not implicit in the mapping method or flux calculation.

The first step addresses the variable spatial coverage of current $pCO_2$ products. Many of the current mapped products only cover roughly 90% of the ocean surface, missing continental shelves and high latitude regions. A newly released global $pCO_2$ climatology product (Landschützer et al. 2020b) includes coverage in the coastal and Arctic regions. We use this climatology to fill any missing areas in each individual product to create consistent full global ocean coverage.

The second methodological step is the choice of flux parameterization, and appropriate scaling of wind speed data. Roobaert et al. (2018) presented uncertainty in air-sea carbon flux induced by various parameterizations of the gas transfer velocity and wind speed data products. Utilizing the MPI-SOMFFN $pCO_2$ product (Landschützer et al. 2020a) and a quadratic

parameterization (Wanninkhof 1992; Ho et al. 2006) they find flux estimates that diverge by 12% depending on the choice of wind speed products. Additionally, they find regional discrepancies to be much more pronounced than global differences, specifically highlighting the equatorial Pacific, Southern Ocean, and North Atlantic as regions most impacted by the choice of wind product. Roobaert et al. (2018) stress that to minimize the uncertainties associated with the wind speed product chosen, the global coefficient of gas transfer must be individually calculated for each (Wanninkhof 1992, 2014). In this work, we assess the impact of wind speed product choice and scaling on six $pCO_2$ products' calculated air-sea flux estimates. By applying a consistent flux calculation methodology to each $pCO_2$ product, we minimize the methodological divergence of fluxes within the ensemble.

Here, we present SeaFlux, a dataset that provides a consistent approach specifically targeting the most commonly used $pCO_2$ data products to deliver an end-product for intercomparisons within assessment studies such as the Global Carbon Budget (Friedlingstein et al. 2020) and the Regional Carbon Cycle Assessment and Processes (RECCAP). The SeaFlux dataset is accompanied by a Python package, called *pySeaFlux* (https://github.com/lukegre/pySeaFlux), that enables users to calculate fluxes for other configurations, use cases and resolutions. Specifically, by first addressing differences in spatial coverage between the observation-based products we can better present a true global $pCO_2$ estimate for each product. SeaFlux also provides gas transfer velocities calibrated to a consistent $^{14}C$ inventory. Further, the data set includes estimates of atmospheric $pCO_2$ and the solubility of $CO_2$ in seawater. Finally, by calculating fluxes using multiple scaled gas transfer velocities for different wind products, we present a methodologically consistent database of air-sea $CO_2$ fluxes calculated from available $pCO_2$ products. SeaFlux is thus an ensemble data product with documented code (pySeaFlux) allowing the community to reproduce consistent flux calculations from various data-based $pCO_2$ reconstructions now and in the future.

## 2. Methods

SeaFlux is based on six observation-based $pCO_2$ products and spans the years 1990-2019 (Table 1). These six products include three neural network derived products (CMEMS-FFNN, MPI-SOMFFN, NIES-FNN), a mixed layer scheme product (JENA-MLS), a multiple linear regression (JMA-MLR), and a machine learning ensemble (CSIR-ML6). These products are included as they have been regularly updated to extend their time period and incorporate additional data that comes with each annual release of the SOCAT database.

All of these methods provide three-dimensional fields (latitude, longitude, time) of the sea surface $pCO_2$ and the air-sea $CO_2$ flux. In their original form, each product may utilize different choices for the inputs to Equation 1 (Table A1). While the choices made by each product's creator, listed in Table A1, are not incorrect, by utilizing a uniform methodology in flux calculation, provided by pySeaFlux, the differences in the resulting flux estimate can be attributed to the $pCO_2$ mapping method itself. In this work we recompute the fluxes using the following inputs to the bulk parameterization approach Equation 1: $k_w$

is the gas transfer velocity (further discussed in Sect. 2.3), *sol* is the solubility of $CO_2$ in seawater, in units mol m$^{-3}$ uatm$^{-1}$, calculated using the formulation by Weiss (1974), near-surface EN4 salinity (Good et al. 2013), NOAA Optimum Interpolation Sea Surface Temperature V2 (OISSTv2) (Reynolds et al. 2002), and European Centre for Medium-Range Weather Forecasts (ECMWF) ERA5 sea level pressure (Hersbach et al. 2020); *ice* is the sea ice fraction from NOAA Optimum Interpolation Sea

Surface Temperature V2 (OISSTv2) (Reynolds et al. 2002); $pCO_2$ is the partial pressure of oceanic $CO_2$ in μatm for each observation-based product after filling, as discussed in Sect. 2.1, and $pCO_2{}^{atm}$ is the dry air mixing ratio of atmospheric $CO_2$ ($xCO_2$) from the ESRL surface marine boundary layer $CO_2$ product available at https://www.esrl.noaa.gov/gmd/ccgg/mbl/data.php (Dlugokencky et al. 2017) multiplied by ERA5 sea level pressure (Hersbach et al. 2020) at monthly resolution, and applying the water vapor correction according to Dickson et al. (2007). All

of the components of Equation 1 are available in the SeaFlux dataset.

Throughout this study, flux is defined as being positive when $CO_2$ is released from the ocean to the atmosphere and negative when $CO_2$ is absorbed by the ocean from the atmosphere. In the following sections, we discuss the three steps that have the greatest impact on the inconsistencies between flux calculations in the six $pCO_2$ products and the approach that we utilize for

the SeaFlux ensemble product.

## 2.1 Step 1: Area filling

Machine learning methods aim to maximize the utility of the existing in situ observations by extrapolation using various proxy variables for processes influencing changes in ocean $pCO_2$. Extrapolation with these independently observed variables is possible due to the nonlinear relationship between $pCO_2$ in the surface ocean and the proxies that drive these changes. However,

not all of the proxy variables have complete global ocean coverage for all months; therefore, the resulting $pCO_2$ products are limited by the extent of the proxy variables (Figure 1, A1). Additionally, in continental shelf regions, there is the potential that different relationships of $pCO_2$ to the proxy variables are expected as opposed to in the open ocean, thus limiting the extrapolations. The mixed layer scheme (utilized by the JENA-MLS product) does not suffer from such missing areas but also does not distinguish between coastal and open ocean; it is stated to be an open-ocean product which is extrapolated to the full

global coverage (Rödenbeck et al. 2013). For this reason, it is not utilized in SeaFlux as a potential product for filling missing areas in the other $pCO_2$ products.

To account for differing area coverage, past studies (Friedlingstein et al. 2019, 2020; Hauck et al. 2020) have adjusted simply by scaling based on the percent of the total ocean area covered by each observation-based product (Figure A2). This does not

account for the fact that some areas have $CO_2$ flux densities that are higher or lower than the global average and their adjustment would be based on that mean value (Table 1,3). Thus, the magnitude of the adjustment by area-scaling is likely an underestimate in some years or products (McKinley et al. 2020). One specific example is the northern high latitudes where

coverage by the six products varies substantially. Similarly, three products provide estimates in marginal seas such as the Mediterranean while the other three products have no reported $pCO_2$ values here.


To address the inconsistent spatial coverage in products we utilize a newly released open and coastal merged climatology product (MPI-ULB-SOMFFN; Landschützer et al. 2020b,c) that is a blend of the coastal ocean SOMFFN mapping method (Laruelle et al. 2017) and the open ocean equivalent (MPI-SOMFFN; Landschützer et al. 2020a). The merged product includes full coverage of open ocean $pCO_2$ along with coastal ocean regions, marginal seas and the Arctic Ocean at 1° by 1° or finer

spatial resolution. A potential alternative resource for filling the missing regions is the OceanSODA-ETHZ surface $pCO_2$ product (Gregor and Gruber, 2021) that maps both the open ocean and marginal seas explicitly for the period 1985-2018 (unlike the JENA-MLS approach). However, as with other products, OceanSODA-ETHZ has limited coverage in the Arctic. A comparison of the overlapping regions between the MPI-ULB-SOMFFN and OceanSODA-ETHZ product shows good agreement (Figure A3). We have confidence moving forward using solely MPI-ULB-SOMFFN for area-filling, as including

OceanSODA-ETHZ would not result in substantially different results and would be constrained by its limited Arctic coverage.

For each observationally-based product, we fill missing grid cells with a scaled value based on the global-coverage MPI-ULB-SOMFFN climatology (Figure 2). The scaling accounts for year-to-year changes in $pCO_2$ in the missing areas (given that the extended MPI-ULB-SOMFFN product is a monthly climatology centered on the year 2006) and is obtained as follows.


To extend the open and coastal merged climatology (MPI-ULB-SOMFFN) to 1990-2019, we calculate a global scaling factor based on the product-based ensemble mean $pCO_2$ for regions that are covered consistently by all six $pCO_2$ products. We first mask all $pCO_2$ products to a common sea mask before taking an ensemble mean ($pCO_2^{ens}$). Next, we divide this ensemble mean by the MPI-ULB-SOMFFN climatology ($pCO_2^{clim}$) at monthly 1° by 1° resolution (Equation 2). The monthly scaling

factor ($sf_{pCO2}$) is calculated by taking the mean over the spatial dimensions. An alternative method of calculating the scaling factor individually for each $pCO_2$ product yields very similar results; the benefit of the ensemble approach is it allows for the scaling factor to be quickly utilized for any other $pCO_2$ product under development.

The scaling factor calculation can be represented as


$$sf_{pCO_2} = mean_{x,y} \left( \frac{pCO_2^{ens}}{pCO_2^{clim}} \right) \qquad (2)$$

where $sf_{pCO_2}$ is the one-dimensional scaling factor (time dimension), $pCO_2^{ens}$ is the ensemble mean of all $pCO_2$ products at three-dimension, monthly 1° by 1° resolution, $pCO_2^{clim}$ is the MPI-ULB-SOMFFN climatology, also at three-dimension but

limited to just one climatological year. The $x$ and $y$ indicate that we take the area-weighted average over longitude ($x$) and

latitude ($y$) resulting in the monthly 1D scaling value. If a product mean is exactly equal to the climatology mean, the scaling factor is 1. The value ranges from 0.91 to 1.06 over the 30-year period. The one-dimensional scaling factor is then multiplied by the MPI-ULB-SOMFFN climatology for each spatial point resulting in a three-dimensional scaled filling map. These values are then used to fill in missing grid cells in each observation-based product.


Globally, the area-filling adjustments result in a difference of less than 17% of the total flux in all products, with the mean adjustment for the six products at 8%. In the Northern Hemisphere, however, the filling process can drive adjustments of up to 32% due to missing coverage in the North Atlantic specifically (Figure 1, Table 3). As expected, the observationally-based products with more complete spatial coverage tend to have smaller flux adjustments. However, the impact on the final $CO_2$

flux depends on both the $\Delta pCO_2$ and wind speed of the areas being filled (Figures 2-3, Table 1,3). The only product that does not change during this adjustment process is the JENA-MLS mixed layer scheme-based product (Rödenbeck et al. 2013) which is produced with full spatial coverage and therefore needs no spatial filling; any difference between filled/unfilled for this product is due to the ocean mask applied in SeaFlux.

Our approach is not without its own assumptions and limitations. We rely on a single estimate to fill the missing $pCO_2$ given that this is the only publicly available coastal-resolution product currently existing. Nevertheless, the fact that common missing areas along coastal regions and marginal seas are reconstructed using specific coastal observations provides a step forward from the linear-scaling approach currently used by the Global Carbon Budget (Friedlingstein et al. 2019, 2020, Figure A2). Further confidence is provided by previous research showing that climatological relevant signals, i.e. mean state and

seasonality, are well reconstructed by the MPI-SOMFFN method (Gloege et al. 2021).

Furthermore, our scaled filling methodology assumes that $pCO_2$ in the missing ocean regions is increasing at the same rate as the common area of open-ocean $pCO_2$ used to calculate the scaling factor. Research from coastal ocean regions and shelf seas reveal that, in spite of a large spatial heterogeneity, this is a reasonable first-order approximation (Laruelle et al. 2018).

Any method of artificially filling in missing areas introduces additional uncertainty to the flux estimates. However, this introduced uncertainty is necessary for true global intercomparison efforts. A concern is that the filling method would artificially lower the spread of the products in the SeaFlux ensemble. We do not find this to be the case. The standard deviation of the mean flux for a most conservative mask, which includes only those grid cells with values reported for all six $pCO_2$ products for all months, is nearly identical to the standard deviation of the final version of the SeaFlux product ensemble. This

comparison indicates that our filling method does not in fact artificially lower the uncertainty or decrease the spread of the products.

## 2.2 Step 2: Wind product selection

Historical wind speed observations (including measurements from satellites and moored buoys) are aggregated and extrapolated through modeling and data assimilation systems to create global wind reanalyses. These reanalyses are required to compute air-sea gas exchange. Air-sea flux is commonly parameterized as a function of the gradient of $CO_2$ between the ocean and the atmosphere with wind speed modulating the rate of the gas exchange (Equation 1). Each of these wind reanalyses has strengths and weaknesses, specifically on regional and seasonal scales (Chaudhuri et al. 2014; Roobaert et al. 2018; Ramon et al. 2019) but all are considered reasonable options by the community (Roobaert et al. 2018). The pySeaFlux package includes options for the user to select their wind product of choice. For the Seaflux ensemble product, we use three wind reanalysis products for completeness: the Cross-Calibrated Multi-Platform v2 (CCMP2, Atlas et al. 2011), the Japanese 55-year Reanalysis (JRA-55, Kobayashi et al. 2015), and the European Centre for Medium-Range Weather Forecasts (ECMWF) ERA5 (Hersbach et al. 2020). The wind speed ($U_{10}$) is calculated at the native resolution of each wind product from u- and v-components of wind. Details of each wind product are shown in Table A2.

## 2.3 Step 3: Calculation of gas transfer

We employ the quadratic wind speed dependence (Wanninkhof 1992; Ho et al. 2006) and calculate the gas transfer velocity ($kw$) for each of the wind reanalysis products as

$$k_w = a \cdot \langle U^2 \rangle \cdot \left(\frac{Sc}{660}\right)^{-0.5} \qquad (3)$$

where the units of $k_w$ are in cm h$^{-1}$, $Sc$ is the dimensionless Schmidt number, and $\langle U^2 \rangle$ denotes the square of average 10-m height winds (m s$^{-1}$), also referred to as the second moment of the wind speed. We choose the quadratic dependence of the gas transfer velocity as it is widely accepted and used in the literature (Wanninkhof 1992; Ho et al. 2006) however we acknowledge that the actual relationship could vary from less than linear (Krakauer et al. 2006) to a cubic (Wanninkhof et al. 1999; Stanley et al. 2009). Observational and modeling studies have often suggested that different parameterizations could be more appropriate under specific conditions such as in regions of high wind speeds (Fairall et al. 2000; Nightingale et al. 2000; McGillis et al. 2001; Krakauer et al. 2006); recent direct carbon dioxide flux measurements made in the high latitude Southern Ocean confirm that even in this high wind environment, a quadratic parameterization fits the observations best (Butterworth & Miller 2016). Future updates of SeaFlux will include $k_w$ for other parameterizations (*e.g. cubic*).

We calculate the square of the wind speed at the native resolution of each wind product and then average to 1° by 1° monthly resolution (see Table A2). The order of this calculation is important, as variability is lost when resampling data to lower resolutions because of the concavity of the quadratic function. For example, taking the square of time-averaged wind speeds would result in an underestimate of the gas transfer velocity (Sarmiento and Gruber 2006; Sweeney et al. 2007). The resulting

second moment is equivalent to $<U^2> = U_{mean}^2 + U_{std}^2$ where $U_{mean}$ and $U_{std}$ are the temporal mean and standard deviation calculated from the native temporal resolution of U.

In addition to the choice of wind parameterization (Roobaert et al. 2018), large differences in flux can result due to the scaling of the coefficient of gas transfer (*a*) applied when calculating the global mean gas transfer velocity. This constant originates from the gas exchange process studies (Krakauer et al. 2006; Sweeney et al. 2007; Müller et al. 2008; Naegler 2009) which utilize observations of radiocarbon data from the GEOSECS and WOCE/JGOFS expeditions (Key et al. 2004). The [14]C released from nuclear bomb testing (hence bomb-[14]C) in the mid-twentieth century has since been taken up by the ocean. The number of bomb-[14]C atoms in the ocean, relative to the pre-bomb [14]C, can thus be used as a constraint on the long-term rate of exchange of carbon between the atmosphere and the ocean. This coefficient, *a*, is not consistent for each wind product and must thus be individually calculated via a cost function that optimizes the coefficient of gas transfer

$$a = k_w \cdot \langle U^2 \rangle^{-1} \cdot \left(\frac{Sc}{660}\right)^{0.5} \qquad (4)$$

where parameters are as defined in Equation 3. The units of the coefficient *a* are $(cm\ h^{-1})\ (m\ s^{-1})^{-2}$. In the cost function, a global average of $k_w$ is set for which several estimates exist in the literature (ranging from 15.1 cm hr[-1] to 18.2 cm hr[-1]), introducing another source of "disharmony" as shown in Table A1 (Krakauer et al. 2006; Naegler et al. 2006; Sweeney et al. 2007; summarised in Table 2 of Naegler et al., 2009). Naegler et al. (2009) show that these estimates fall within the ~20% range of uncertainty of the bomb-[14]C constrained global average $k_w$, which he estimates at $16.5 \pm 3.2$ cm hr[-1]. We scale $k_w$ to this single value (16.5 cm hr[-1]) over the three-decade period 1990-2019.

Our scaled coefficients (Table 2) correspond well with the estimate of Wanninkhof (2014) who uses the CCMP wind product to estimate *a* as 0.251, where our estimate of *a* for CCMP is 0.257. Scaling $k_w$ to a single global value (here, 16.5 cm hr[-1]) for all wind products reduces the spread of flux estimates, but it does not reduce the uncertainty which remains ~20%. This uncertainty must be accounted for when reporting fluxes (Naegler 2009; Wanninkhof 2014). In this work, we refer to this uncertainty, which is inherent to the formulation and scaling of $k_w$, as intrinsic uncertainty, which we do not try to reduce with SeaFlux and include in our reported uncertainty estimate. However, by correctly scaling $k_w$ for each wind product we reduce the disharmony associated with incorrect scaling by up to 9%, depending on which $pCO_2$ and wind reanalysis product are considered. This is consistent with previous results shown by Roobaert et al. (2018, 2019).

## 2.4 Further parameters for flux calculation

The remaining parameters of Equation 1 are the solubility of $CO_2$ in seawater (*sol*), the atmospheric partial pressure of $CO_2$ ($pCO_2^{atm}$), and the area weighting to account for sea ice cover. While the choices of products used for these parameters can also result in differences in flux estimates, the impacts are much smaller as compared with the parameters discussed above.

Atmospheric $pCO_2$ is calculated as the product of surface $xCO_2$ and sea level pressure corrected for the contribution of water vapor pressure. The choice of the sea level pressure product or absence of the water vapor correction can have a small, but not insignificant, impact on the calculated fluxes. Additionally, some products utilize the output of an atmospheric $CO_2$ inversion product (*e.g.* CarboScope, Rödenbeck et al. 2013; CAMS $CO_2$ inversion, Chevallier, 2013) which can introduce differences in the flux estimate outside of the sources related to a product's surface ocean $pCO_2$ mapping method. Importantly, we do not advocate that our estimate of $pCO_2^{atm}$ is an improvement over other estimates thereof; rather we provide an estimate of $pCO_2^{atm}$ that has few assumptions and leads to a methodologically consistent estimate of $\Delta pCO_2$. We maintain the same philosophy in our estimates of solubility of $CO_2$ in seawater and sea-ice area weighting and therefore we do not elaborate on them here.

## 3. Results and Discussion

### 3.1 SeaFlux air-sea $CO_2$ flux calculation

Following Equation 1, $CO_2$ flux is calculated individually for each of the six observation-based products with three available wind products (CCMPv2, ERA5, JRA55) as discussed in Sect. 2.2 (Table 4). Since we account for spatial coverage differences via our filling method (Sect. 2.1), taking a global mean flux for each of the data products truly follows the definition of "global" for each original product. Figure 4 shows the difference these wind products generate on the resulting global mean flux of the CSIR-ML6 product as one example (other products in Figure A4). The three wind products show very consistent fluxes throughout the time series, however, the importance of appropriate scaling of the coefficient of gas transfer (*a*) is evident by the significant differences between global mean fluxes calculated with unscaled and scaled *a* values (Figure 4, Table 2). It is clear that the impact of applying the appropriate coefficient of gas transfer through proper scaling has a larger impact on the resulting flux time series than solely the choice of wind product.

### 3.2 SeaFlux ensemble flux

By calculating each product's air-sea $CO_2$ flux using consistent inputs described in Section 3.1, we permit for a more accurate comparison of fluxes with the SeaFlux ensemble. Combining all fluxes, we derive a mean flux estimate of -1.97 $\pm$ 0.45 PgC yr$^{-1}$ (Table 4). We discuss the calculation of the uncertainty in the following section. This flux estimate is strengthened by the use of multiple observation-based $pCO_2$ products and wind products which we consider to be independent estimates for the purpose of the uncertainty calculation. These flux values are different from those produced by the observation-based $pCO_2$

product's original creator, both spatially and on the mean (Figure 5, A5, Table A1, A3). However, by calculating fluxes in such a consistent manner, we on the one hand gain more confidence in the ensemble mean estimate as it considers representations using a variety of pCO$_2$ reconstructions, gas transfer parametrizations and wind products, and on the other hand, we have a more realistic uncertainty representation than previous estimates based on a single pCO$_2$ reconstruction.

### 3.3 Uncertainty discussion

All flux estimates using such parameterizations are not without significant uncertainties and SeaFlux is no exception. We estimate the uncertainty of the flux estimate to be 0.45 PgC yr$^{-1}$. Here, the stated spread represents $\sqrt{\sum(\sigma_{wind}^2, \sigma_{pco2}^2, \sigma_{kw}^2)}$ where $\sigma_{pco2}$ (0.19 PgC yr$^{-1}$) is the mean standard deviation over the six filled pCO$_2$ products and $\sigma_{wind}$ (0.09 PgC yr$^{-1}$) is the mean standard deviation over the three wind products included in the SeaFlux product. $\sigma_{kw}^2$ (0.39 PgC yr$^{-1}$) is the 20% uncertainty in the gas transfer velocity and associated scaling flux parameterization (Wanninkhof 2014). This last estimate shows that there is significant intrinsic uncertainty inherent to the method of calculation as estimated by Naegler (2009) and Wanninkhof (2014).

Currently, there is only one product available designed to estimate the pCO$_2$ of coastal oceans, the Arctic Ocean and marginal seas. It would be beneficial to likewise have an ensemble of estimates in these regions to better constrain the uncertainty attached to this filling approach. Therefore, while our current analysis shows that the chosen filling method does not itself reduce the spread in the products, we push the community to extend their products to the coastal ocean so as to eliminate the need for this correction in the future.

While the SeaFlux product is unable to further reduce these sources of uncertainty, the strength of the product is that it provides an estimated flux with no source of difference that is not implicit in the mapping method or flux calculation.

### 3.4 Issues not addressed by SeaFlux

While SeaFlux presents one approach to standardize the calculation of air-sea carbon flux, there remain issues that the ocean carbon community is still working towards understanding and incorporating. One such issue has been raised by Watson et al. (2020) who contend that a correction should be applied to in situ pCO$_2$ observations to account for the vertical temperature gradient between the ship water intake depth and the surface skin layer where gas exchange actually takes place. A further correction should be applied when calculating fluxes to account for the "cool skin" effect caused by evaporation (Woolf et al. 2016; Watson et al. 2020). Applying these corrections results in an increasing CO$_2$ sink by up to 0.9 PgC yr$^{-1}$ (Watson et al. 2020). Here, we do not take such adjustments into account for two reasons. Firstly, the skin temperature correction to pCO$_2$ needs to be applied directly to the measurements and not the final interpolated pCO$_2$ from the data products. Hence, it is up to the developers of the SOCAT dataset and the developers of the pCO$_2$ mapping products to decide on the inclusion of this

correction. It would then be up to the developers of the data products to update their mapped products. Secondly, the cool skin correction would be equally applied to all methods and would not contribute to the inconsistencies in flux calculation that we are trying to address here. As the ocean carbon community moves towards consensus on such issues, SeaFlux will be updated to include revised protocols.


To compare these estimates of contemporary air-sea net flux (Fnet) from surface ocean $pCO_2$ with estimates of the anthropogenic carbon flux (Fant) from interior data (Mikaloff Fletcher et al. 2006; DeVries 2014; Gruber et al. 2019), or from global ocean biogeochemical models (Friedlingstein et al. 2020; Hauck et al. 2020), it is necessary to account for the outgassing of natural carbon, which was supplied to the ocean by rivers, as well as the non-steady-state behavior of the natural carbon

cycle (Hauck et al. 2020). Work is ongoing to quantify the lateral river carbon flux transported into the coastal and open oceans. Current estimates are 0.23 PgC yr$^{-1}$ (Lacroix et al. 2020), 0.45 PgC yr$^{-1}$ (Jacobsen et al. 2007), and 0.78 PgC yr$^{-1}$ (Resplandy et al. 2018), and the regional distribution of the resulting outgassing remains understood only from a few model simulations (Aumont et al. 2001; Lacroix et al. 2020). In addition, quantification of non-steady-state behavior of the natural carbon cycle has only recently been proposed and significant uncertainty remains, with a magnitude range of 0.05-0.4 PgC yr$^{-1}$ for 1994-

2007 (Gruber et al. 2019; McKinley et al. 2020). Similar to the "cool skin" correction suggested by Watson et al. (2020) discussed above, in this work we have not included these adjustments here as they would not contribute to the inconsistencies between the different products for Fnet itself, which is our focus.

## 4. Conclusions

We introduce SeaFlux, a data set that facilitates a standardized approach for flux calculations from observationally-based $pCO_2$

products. Specifically, we address the two largest sources of divergence, namely the differences in spatial coverage between the products, and the scaling of the gas transfer velocity for available wind speed products based on global [14]C-based constraints. The area adjustment is the largest contributor to the methodological discrepancies, resulting in an increase in $CO_2$ uptake of 0-17% relative to the original, possibly incomplete coverage (depending on $pCO_2$ product). The global scaling of the gas transfer velocity can change the $CO_2$ flux on average by 5% relative to non-standardized flux calculations. The impact

of applying the appropriate gas exchange coefficient through proper scaling has a larger impact on the resulting flux time series than solely the choice of wind product. By accounting for these sources of differences, the global mean calculated air-sea carbon flux calculated from the six available products is adjusted by up to 21%. The SeaFlux ensemble mean air-sea carbon flux is estimated to be -1.97 +/- 0.45 PgC yr$^{-1}$ with the spread representing 1σ as calculated from the 18 realizations.

This work provides an ensemble data product of the sea-air $CO_2$ flux based on observation-based $pCO_2$ products. This ensemble product is meant to facilitate the use of the $pCO_2$ observation-based ocean flux estimates in assessment studies of the global carbon cycle, such as the Global Carbon Budget or RECCAP-2. Note that the original $pCO_2$ products still offer additional

information important in other applications, such as coverage over longer time periods, higher spatial or temporal resolution, or runs incorporating further auxiliary data sets or $pCO_2$ data (e.g., SOCCOM float data, Bushinsky et al. 2019).


Along with the ensemble of $CO_2$ flux fields, we also provide a public-use coding package (pySeaFlux) allowing users to apply the presented standardized flux calculations to their own data-based $pCO_2$ reconstructions.

**Data and Code Availability**

Data (Gregor & Fay 2021) is available on Zenodo (https://doi.org/10.5281/zenodo.5148795) and the software used to generate
this data is available on GitHub (https://github.com/lukegre/pySeaFlux). NOAA_OI_SST_V2 data provided by the NOAA/OAR/ESRL PSL, Boulder, Colorado, USA, from their Web site at https://psl.noaa.gov/data/gridded/data.noaa.oisst.v2.html.

**Author Contributions**

ARF and LG designed the experiment and LG developed the model code and performed the simulations with ARF focusing on analysis. ARF and LG collectively prepared the manuscript with contributions from all co-authors.

**Acknowledgements**

P.L, N.G and L.G received funding from the European Community's Horizon 2020 Project under grant agreement no. 821003
(4C). GAM and ARF received funding from Columbia University and the National Science Foundation OCE1948624; GAM was also supported by National Oceanic and Atmospheric Administration agreement NA20OAR4310340. The Surface Ocean $CO_2$ Atlas (SOCAT) is an international effort, endorsed by the International Ocean Carbon Coordination Project (IOCCP), the Surface Ocean Lower Atmosphere Study (SOLAS) and the Integrated Marine Biosphere Research (IMBeR) program, to deliver a uniformly quality-controlled surface ocean $CO_2$ database. The many researchers and funding agencies responsible
for the collection of data and quality control are thanked for their contributions to SOCAT.

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

**Figures**

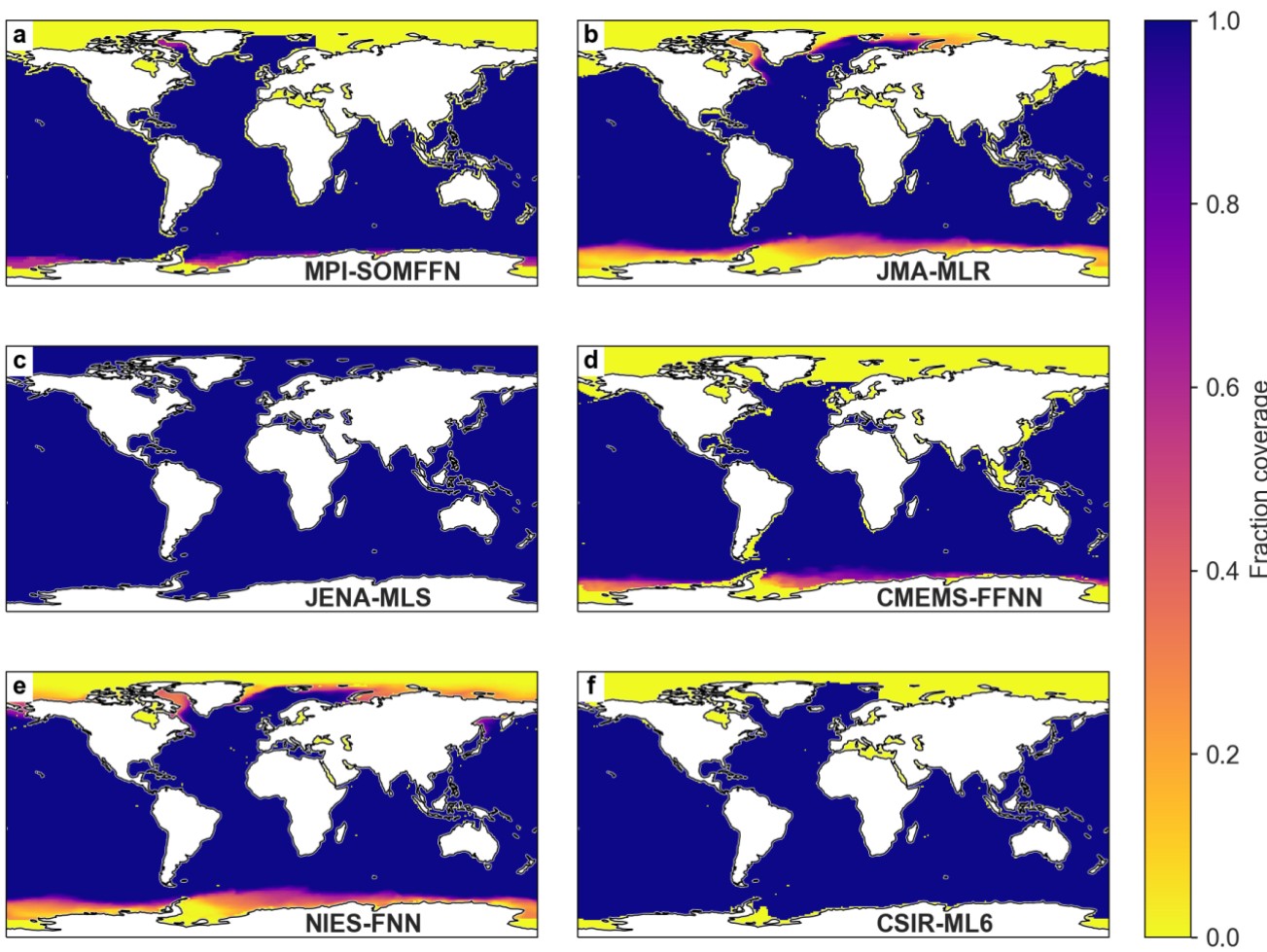

**Figure 1:** Maps showing the fraction of months (1990-2019) with coverage available for each of the six $pCO_2$ data products used in this study. Blue regions represent full temporal coverage of $pCO_2$ in the product while yellow areas show regions with no reported $pCO_2$ values for any month of the time series.


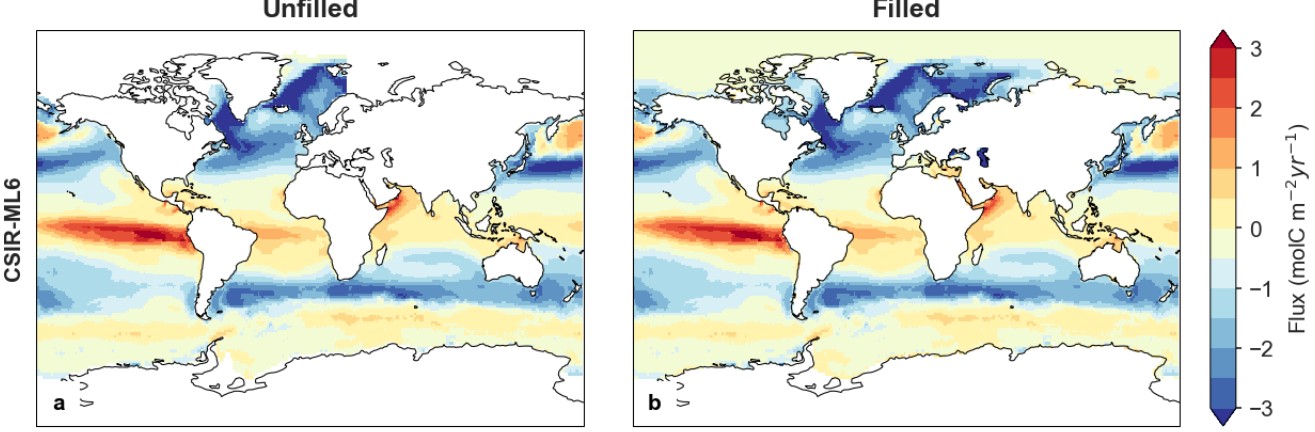

**Figure 2:** Maps demonstrating the filling procedure employed in this study using a snapshot of $pCO_2$ from May 2013. (a) map of unfilled CSIR-ML6 $pCO_2$. (b) the scaled $pCO_2$ climatology of Landschützer et al. (2020b). (c) the mean $pCO_2$ for the scaled climatology over time. (d) the CSIR-ML6 $pCO_2$ product (a) filled using the scaled climatology (b).


**Figure 3:** Mean flux (mol m$^{-2}$ yr$^{-1}$), 1990-2019, for CSIR-ML6 product. (a) map of mean calculated flux using the original $pCO_2$ product and 3 scaled wind products (CCMPv2, ERA5, JRA55); (b) map of mean calculated flux using the filled $pCO_2$ product and 3 scaled wind products. Similar maps for all other products are available in Figure A5.


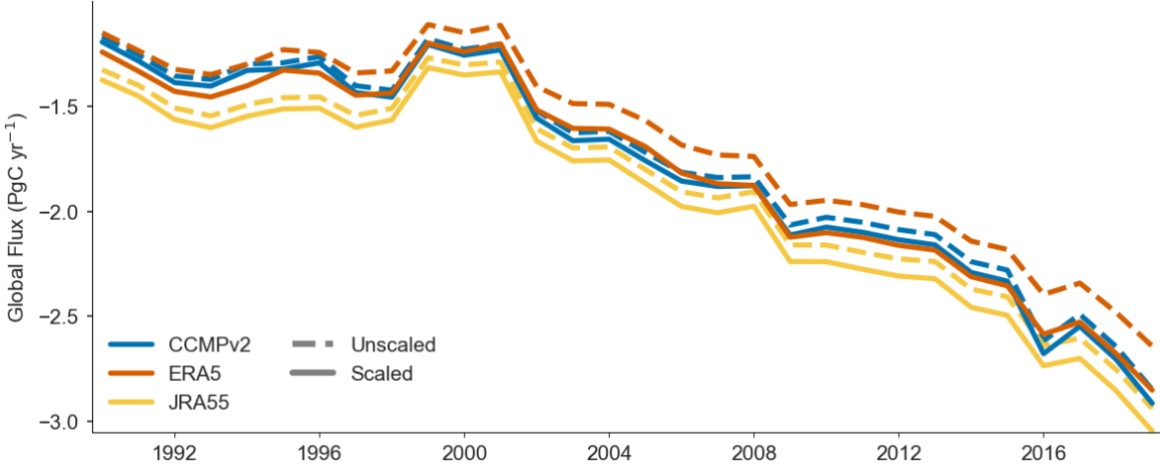

**Figure 4:** CSIR-ML6 product calculated air-sea $CO_2$ flux time series for various wind speed products; scaled (solid) and unscaled (dashed; $a = 0.251$). Time series plots for all $pCO_2$ products and including 2 additional wind products (NCEP1 and NCEP2) are included in Figure A4.

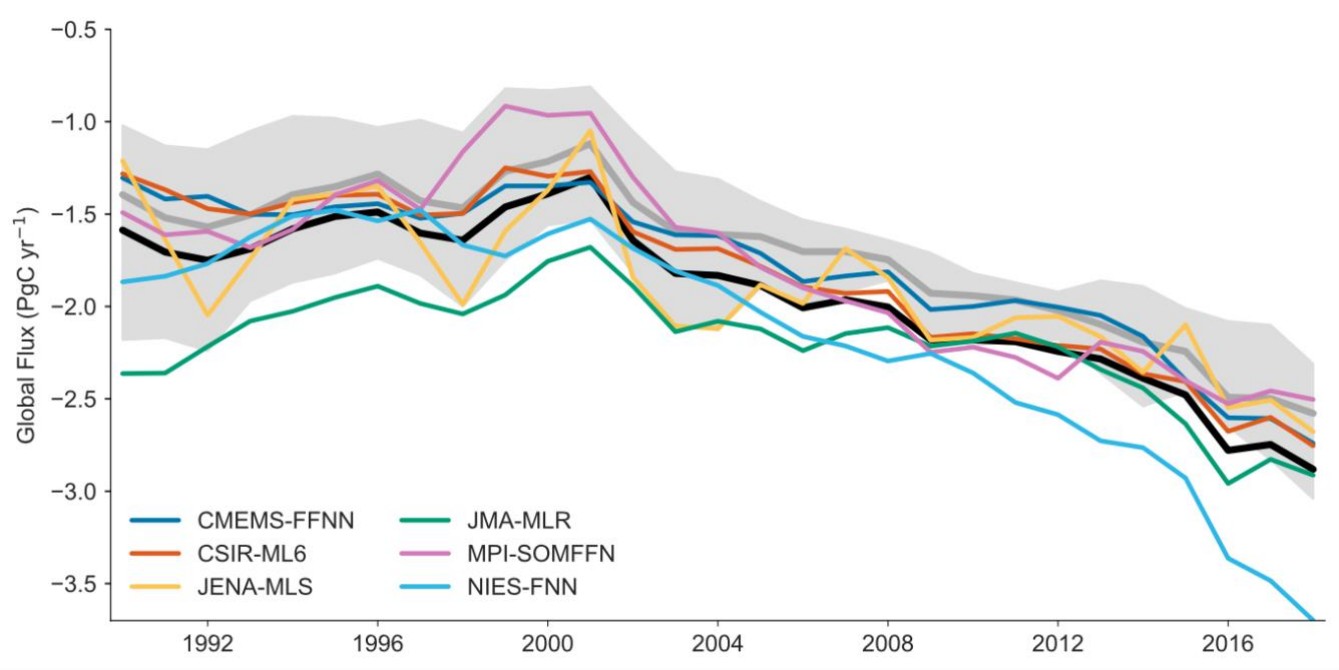

**Figure 5:** Global flux time series from six observation-based products. Color lines show fluxes calculated from the standardized approach presented here (spatial filling with flux calculated from three wind products and the average flux is then plotted here); the black line shows the mean of six products. The shaded region shows the spread of original flux calculations from product creators with the mean in gray.

## Tables

**Table 1:** Global area coverage and mean $pCO_2$ for the six observation-based products. Area coverage listed represents the average annual area covered for 1990-2019 as this value changes monthly for many products (Figure A1). Change is defined as the filled product – original product (i.e. a negative change implies the original product had a larger global/regional mean $pCO_2$ than the filled product). Global and hemispheric mean $pCO_2$ value for filled/original coverage is included in parentheses below the delta value.

| Product | Area coverage (% global ocean) | Mean Global $pCO_2$ change (µatm) | Northern Hem $pCO_2$ change (µatm) | Southern Hem $pCO_2$ change (µatm) |
|---|---|---|---|---|
| CMEMS-FFNN *Denvil-Sommer et al. 2019 Chau et al. 2020* | 89% | -1.50 (364.86/366.36) | -3.96 (362.86/366.81) | 0.33 (366.36/366.03) |
| CSIR-ML6 *Gregor et al. 2019* | 93% | -0.75 (364.23/364.98) | -1.72 (362.10/363.82) | 0.07 (365.81/365.74) |
| JENA-MLS *Rödenbeck et al. 2013* | 100% | 0.00 (362.35/362.35) | 0.00 (357.87/357.87) | 0.00 (365.70/365.70) |
| JMA-MLR *Iida et al. 2020* | 85% | -0.50 (362.45/362.95) | -1.97 (360.02/361.98) | 0.77 (364.26/363.49) |
| MPI-SOMFFN *Landschützer et al. 2014 Landschützer et al. 2020a* | 89% | -0.90 (364.61/365.50) | -2.18 (362.50/364.68) | 0.17 (366.18/366.01) |
| NIES-FNN *Zeng et al. 2014* | 91% | -0.23 (361.56/361.80) | -0.86 (360.75/361.62) | 0.25 (362.16/361.91) |

**Table 2: CSIR-ML6 product flux values** Column 1 lists the scaled coefficient of gas transfer for each of the 3 wind reanalysis products; column 2 included the global mean flux using each wind product. Column 3 shows the difference in resulting flux when using a scaled coefficient of gas transfer versus a set value of 0.26. All flux values reported are from the area-filled product version. All values are computed over the period 1990-2019.

| Wind product | Scaled coefficient of gas transfer ($a$) | Global flux mean (PgC yr$^{-1}$) | Mean flux difference scaled – unscaled |
|---|---|---|---|
| CCMP2 | 0.257 | -1.81 | -0.04 |
| ERA5 | 0.271 | -1.81 | -0.13 |
| JRA55 | 0.260 | -1.96 | -0.07 |

**Table 3:** Mean air-sea fluxes (PgC yr$^{-1}$), 1990-2019, using the mean of three wind products, calculated for the filled global area and the unfilled native "global" area for each $pCO_2$ product. The northern hemisphere (NH) and southern hemisphere (SH) fluxes (unfilled/filled) are included to highlight the imbalanced regional effect of the spatial filling process.

| Product | Global Flux unfilled/filled | NH Flux unfilled/filled | SH Flux unfilled/filled |
|---|---|---|---|
| CMEMS-FFNN | -1.50/-1.82 | -0.62/-0.91 | -0.88/-0.91 |
| CSIR-ML6 | -1.74/-1.86 | -0.82/-0.93 | -0.92/-0.93 |
| JENA-MLS | -1.91/-1.91 | -0.91/-0.91 | -0.99/-0.99 |
| JMA-MLR | -2.00/-2.23 | -0.94/-1.15 | -1.06/-1.08 |
| MPI-SOMFFN | -1.61/-1.81 | -0.75/-0.93 | -0.86/-0.88 |
| NIES-FNN | -2.16/-2.21 | -0.88/-0.92 | -1.28/-1.28 |

**Table 4:** Mean fluxes (PgC yr$^{-1}$) for each observational pCO$_2$ product over the period 1990-2019. Mean flux calculated from filled coverage pCO$_2$ map and scaled gas exchange coefficient; global mean flux is for 3 wind products (CCMP2, ERA5, JRA55) and the average. The time series of the mean flux values for each product (rightmost column) are plotted in Figure 5.

| pCO$_2$ mapping Product | CCMPv2 | ERA5 | JRA55 | MEAN |
|---|---|---|---|---|
| CMEMS-FFNN | -1.77 | -1.77 | -1.92 | -1.82 $\pm$ 0.09 |
| CSIR-ML6 | -1.81 | -1.81 | -1.96 | -1.86 $\pm$ 0.08 |
| JENA-MLS | -1.86 | -1.85 | -2.01 | -1.91 $\pm$ 0.10 |
| JMA-MLR | -2.18 | -2.18 | -2.34 | -2.23 $\pm$ 0.09 |
| MPI-SOMFFN | -1.77 | -1.76 | -1.91 | -1.81 $\pm$ 0.09 |
| NIES-FNN | -2.15 | -2.17 | -2.30 | -2.21 $\pm$ 0.08 |
| **MEAN** | -1.92 $\pm$ 0.19 | -1.92 $\pm$ 0.20 | -2.07 $\pm$ 0.19 | -1.97 $\pm$ 0.21 |

 **Appendix A**

**Table A1:** Summary of parameters used to calculate flux

| pCO$_2$ mapping Product | Wind speed product | Scaling of gas transfer value | Atmos surf pressure | Gas exchange Parameterization |
|---|---|---|---|---|
| This study | Calculated for three and final result is an average of the resulting fluxes: ERA5, JRA55, CCMP2 | Scaled to 16.5 cm/hr | ERA5 Hersbach et al (2020) | Quadratic Wanninkhof (1992) |
| CMEMS-FFNN *Denvil-Sommer et al. 2019; Chau et al. 2020* | ERA5 Hersbach et al (2020) | Scaled to 16.0 cm/hr | CAMS inversion Chevallier (2013) | Quadratic Wanninkhof (1992) |
| CSIR-ML6 *Gregor et al. 2019* | ERA5 Hersbach et al (2020) | Scaled to 16.0 cm/hr | ERA5 Hersbach et al (2020) | Quadratic Wanninkhof (1992) |
| JENA-MLS *Rödenbeck et al. 2013* | NCEP1 Kalnay et al (1996) | Scaled to 16.5 cm/hr | NCEP1 Kalnay et al (1996) | Quadratic Wanninkhof (1992) |
| JMA-MLR *Iida et al. 2020* | JRA55 Kobayashi et al. (2015) | Scaled to 16.5 cm/hr | JRA55 Kobayashi et al. (2015) | Quadratic Wanninkhof (1992) |
| MPI-SOMFFN *Landschützer et al. 2020a* | ERA5 Hersbach et al (2020) | Scaled to 16.0 cm/hr | NCEP1 Kalnay et al. (1996) | Quadratic Wanninkhof (1992) |
| NIES-FNN *Zeng et al. 2015* | NCEP1 Kalnay et al. (1996) | Utilized $a = 0.26$ Takahashi et al. (2009) | NCEP1 Kalnay et al. (1996) | Quadratic Wanninkhof (1992) |

**Table A2:** Summary of wind products used in this study. Note that the date range starts for the first full year of data. We do not use NCEP1/2 in our main results, but these are included for reference. Time units are in hours and space in degrees. Mean wind speed is given for the ice-free ocean for the three-decade period 1990-2019.

| Product name | Resolution | | Date range | Mean speed (m s$^{-1}$) | Scaling ($a$) | Reference |
|---|---|---|---|---|---|---|
| | Time | Space | | | | |
| Cross-Calibrated Multi-Platform v2 | 6 | 0.25 | 1988-present | 7.7 | 0.257 | Atlas et al. (2011) |
| ECMWF Reanalysis 5th Generation | 1 | 0.25 | 1979-present | 7.5 | 0.271 | Hersbach et al. (2020) |
| Japanese 55-year Reanalysis | 3 | 0.50 | 1958-present | 7.6 | 0.260 | Kobayashi et al. (2015) |
| NCEP-NCAR reanalysis 1 | 6 | 2.50 | 1948-present | 7.2 | 0.287 | Kalnay et al. (1996) |
| NCEP-NCAR reanalysis 2 | 6 | 2.50 | 1979-present | 8.3 | 0.218 | Kanamitsu et al. (2002) |

**Table A3:** Mean fluxes, PgC yr$^{-1}$, 1990-2019 for each observational pCO$_2$ product. Mean flux calculated from unfilled (filled) coverage pCO$_2$ map and unscaled/scaled coefficient of gas transfer (unscaled = 0.251); calculated for 3 wind products (CCMP2, ERA5, JRA55) with the average shown here. Percent change is calculated as the difference between the unfilled/unscaled and filled/scaled as a fraction of the filled/scaled; does not indicate an error in the product's flux but is a representation of the impact the filling and scaling can have on the end flux estimate. The mean flux as reported in the original pCO$_2$ product is included for comparison (Figure 5).

| pCO$_2$ mapping Product | Unfilled, Unscaled | Filled, Scaled | % Change | Original product |
|---|---|---|---|---|
| CMEMS-FFNN | -1.44 | -1.82 | 21% | -1.75 |
| CSIR-ML6 | -1.66 | -1.86 | 11% | -1.55 |
| JENA-MLS | -1.82 | -1.91 | 5% | -1.93 |
| JMA-MLR | -1.91 | -2.23 | 15% | -1.99 |
| MPI-SOMFFN | -1.54 | -1.81 | 16% | -1.49 |
| NIES-FNN | -2.06 | -2.21 | 7% | -1.61 |


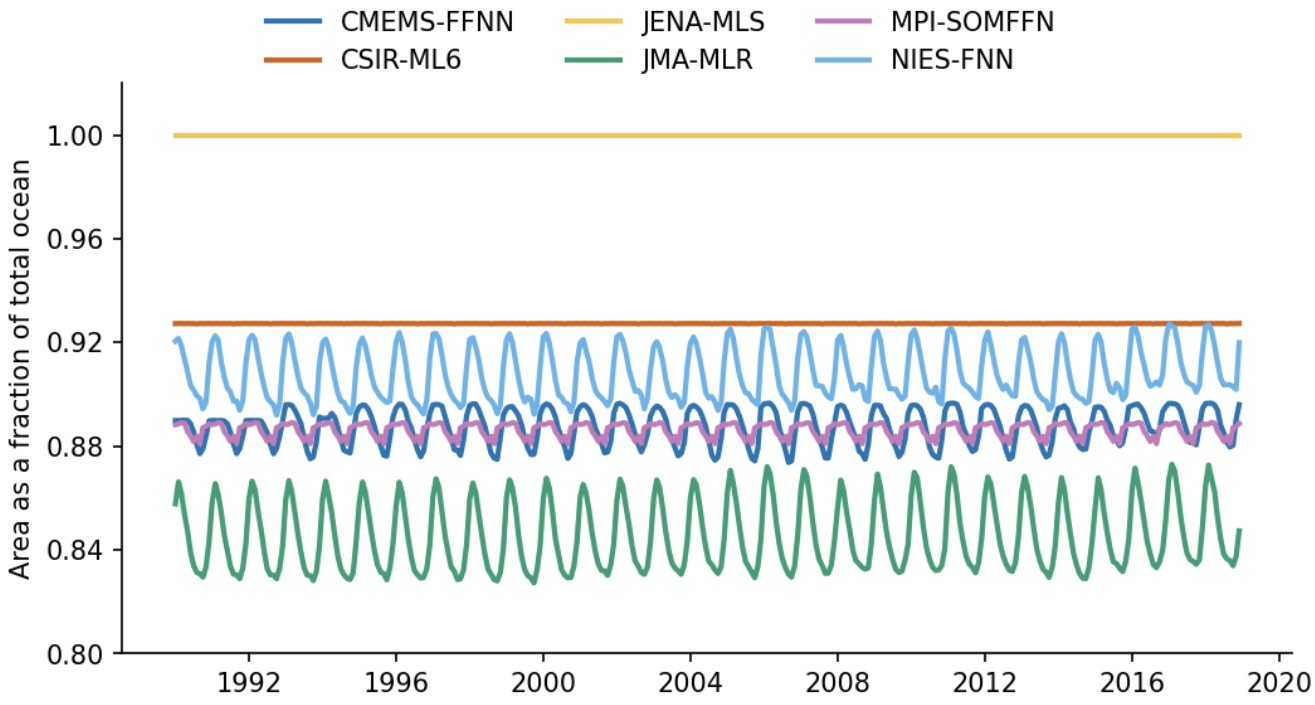

**Figure A1:** Time series showing the fraction of area covered by observations as a function of time (monthly) for the six pCO$_2$ data products used in this study.

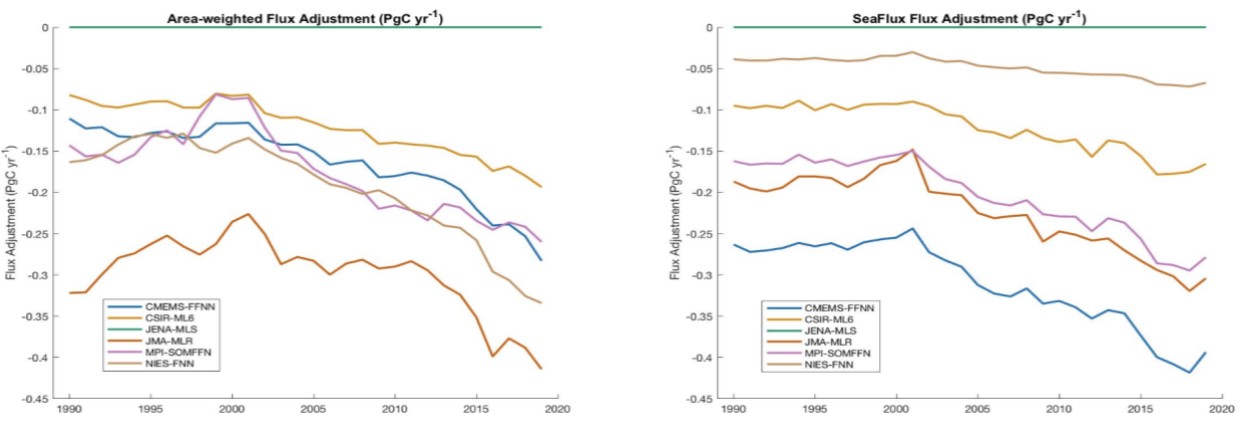

**Figure A2:** Annual time series of the additional flux amount calculated by the area-weighted method used in the Global Carbon Budget (a) and a similar plot showing the annual additional flux using the SeaFlux methodology (b).

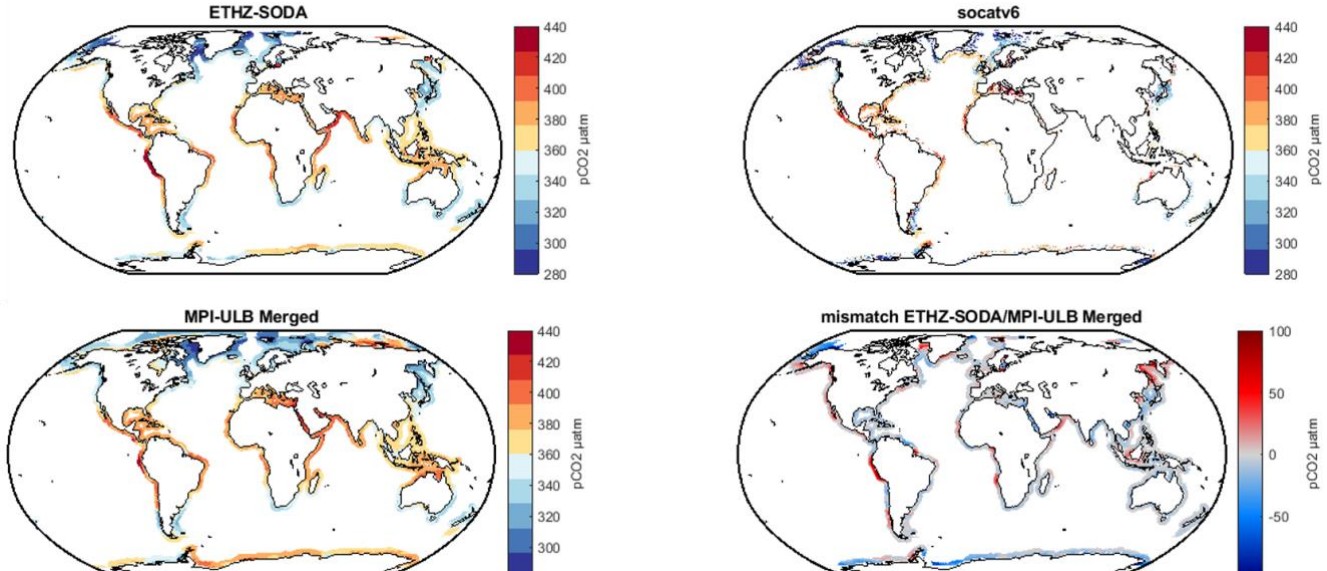

**Figure A3:** Spatial distributions of the annual mean pCO$_2$ (µatm) generated by (a) ETHZ-OceanSODA, (b) extracted from the SOCATv6 database and (c) from the MPI-ULB Merged product. (d) bias between panels (a) and (c) in µatm (red colors correspond to regions in which the pCO$_2$ from ETHZ-OceanSODA is higher than MPI-ULB Merged product). There is good agreement between the products on a regional scale.

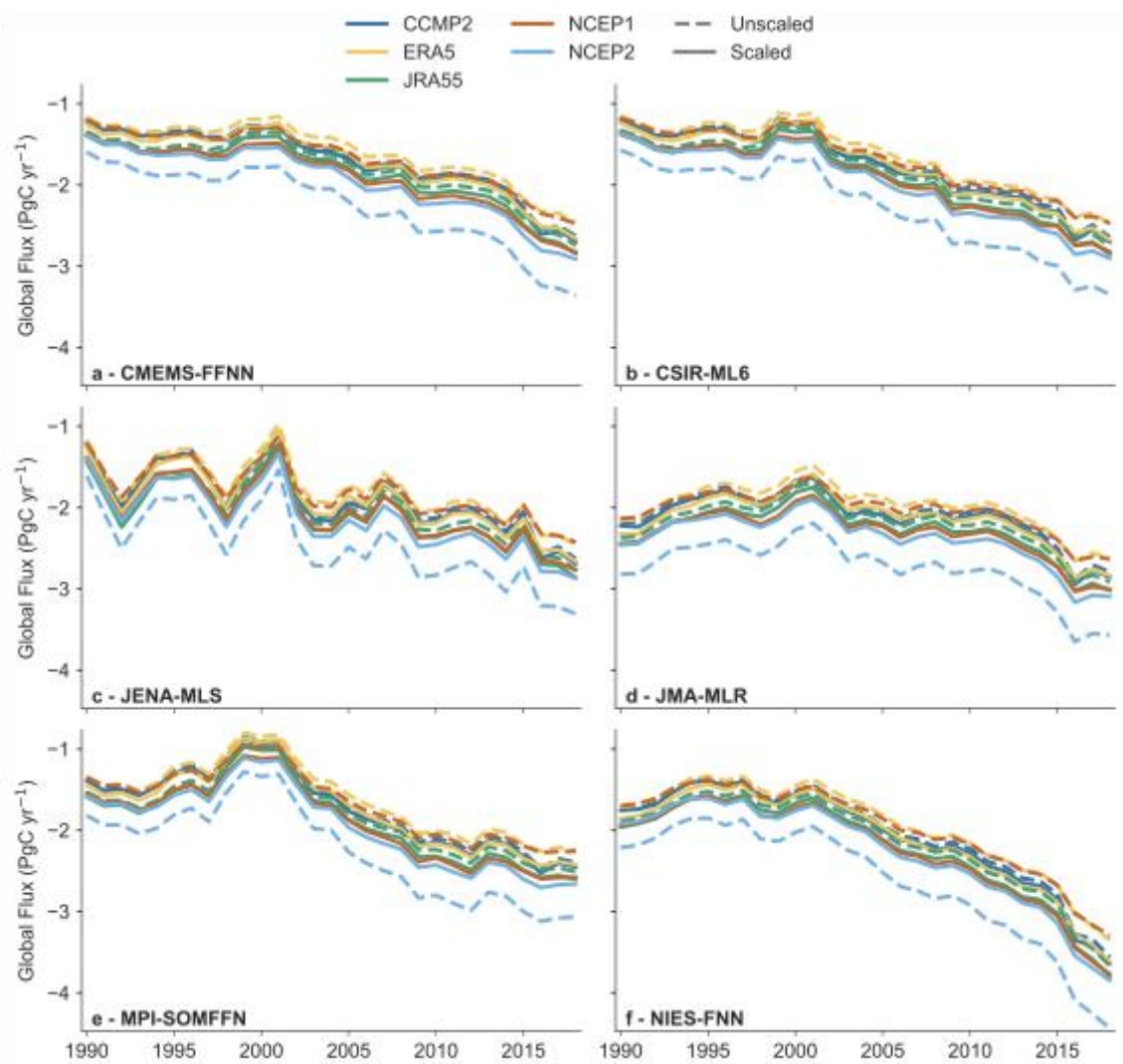

**Figure A4:** Air-sea CO$_2$ flux time series (PgC yr$^{-1}$) calculated using five wind speed products (CCMPv2, ERA5, JRA55, NCEP1, NCEP2); scaled (solid) and unscaled (dashed).

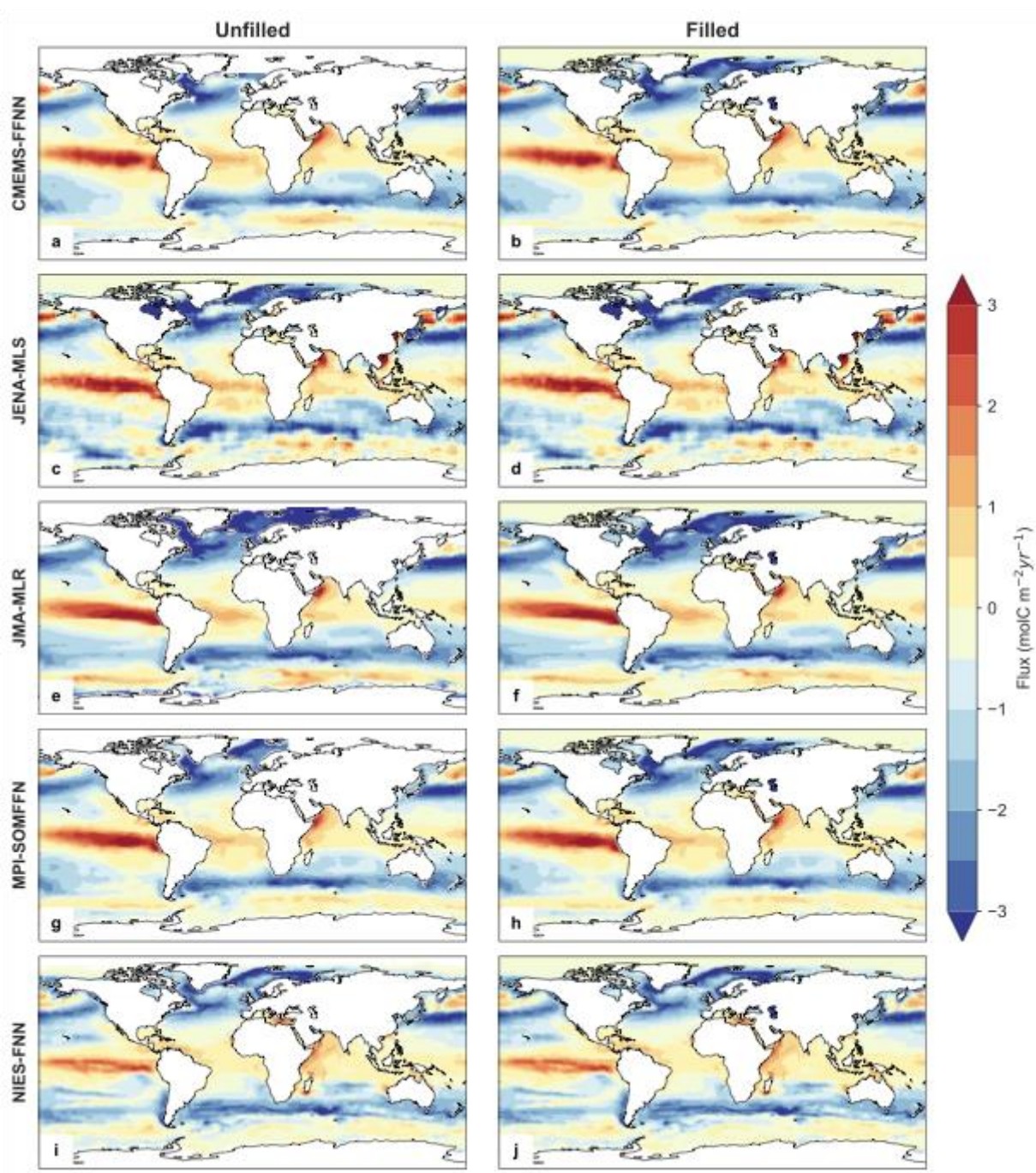

**Figure A5:** Mean flux (mol m$^{-2}$ yr$^{-1}$), 1990-2019. Left-hand column: map of mean calculated flux using the unfilled pCO$_2$ product and 3 scaled wind products. Right-hand column: map of mean calculated flux using the filled pCO$_2$ product and 3 scaled wind products.