# Peer review of "SeaFlux: harmonization of air-sea CO2 fluxes from surface pCO2 data"

_Earth System Science Data, 2021_

## Author Comment (AC1)

**Reviewer Rik Wanninkhof NOAA/AOML**

This is a well written manuscript thoughtfully describes and rectifies inconsistencies between several global ocean air-sea CO2 flux products. Notably, differences in ocean surface area depending if marginal and coastal seas are included; and providing an adjustment to quadratic gas transfer parameterizations to match the global bomb 14C constraint. This "harmonization" leads to a better agreement between the different products. As a last step the authors use an ensemble approach to determine a global mean air sea CO2 flux of 1.92 +-0.35 Pg C/yr where the uncertainty is a 2 sigma (95 %) confidence interval and is based on agreement of the 18-member ensemble using on 6 interpolated and area normalized surface water pCO2 products and three different windspeed products with normalized gas transfer velocities. The uncertainty (+-0.17) (1-sigma) basically reflects the differences in interpolation techniques of the same dataset (SOCAT). Using this uncertainty for purposes as stated "help to identify missing fluxes as we strive to close the global carbon budget" is not appropriate. While beyond the scope of ESSD, it looks like the authors are trying "to sneak a number" into the peer reviewed literature.

The manuscript appears a mixture of a data description as expected for ESSD and commentary/interpretation which is beyond the scope of this journal. As listed on the home page of ESSD: "for the publication of articles on original research data (sets),. The editors encourage submissions on original data or data collections which are of sufficient quality and have the potential to contribute to these aims. Any interpretation of data is outside the scope of regular articles."

*Response: Many thanks for the overall positive assessment of our study and the helpful comments. We have edited the manuscript to be inline with the scope of the ESSD journal, specifically focusing on the novel aspect of the computational tools provided by the pySeaFlux package. While we do still include the mean air-sea CO2 flux estimate from this group of products in the manuscript text we have removed it from the abstract. We have also focused on the uncertainty quantification at each step in the pySeaFlux package, including uncertainties inherent to the observations themselves, and elaborate on such in the text. We have added an additional paragraph to the Area-filling discussion (Section 2.1) that specifically reports estimates of uncertainty for this method. Additionally, we have added a section to the Results and Discussion section that outlines various types of uncertainty: the intrinsic uncertainty, introduced uncertainty, and the spread of the resulting ensemble of products.*

**R1: Why isn't the "Landschutzer, P., Laruelle, G., Roobaert, A., and Regnier, P.: A combined global ocean pCO2 climatology combining open ocean and coastal areas (NCEI Accession 0209633)," included as one of the interpolated products? This seems to be one of the most complete sets wrt area. Also, citing the paper in ESSD rather than the data product might be more appropriate.**

*Response: The Landschützer et al 2020 climatology is not included as an interpolated product because it is limited in its temporal coverage- it is a 12 month climatology rather than a time-evolving product. We have edited the manuscript to reference both the ESSD manuscript and the data product itself.*

**R1: The authors refer to the "Seaflux product and "Seaflux package" but the full description of either is lacking in this paper. The Gregor and Fay, 2021 referenced is the dataset without a complete description. What exactly is the Seaflux package? I would encourage the authors to focus on describing this product and tools in this manuscript.**

*Response: Thank you for this comment and suggestion. To clarify this, we have decided to call the Python package pySeaFlux, which contains all the code to calculate air-sea CO2 flux from pCO2. The SeaFlux product (the subject of this manuscript) is the ensemble of the 6 pCO2 products with area-filling to produce global coverage. This product will therefore be evolving as more products become available.*

**R1: It is mentioned that the area normalization has been previously applied in models and products in a rudimentary fashion. How different are the global fluxes using simple extrapolation methods compared to the approach used here? Eyeballing the results it appears that scaling global fluxes to a consistent area, and has been done it the past seems to work reasonably well.**

*Response: The additional flux for each product resulting from the area-filling method proposed here in the pySeaFlux package is on the same scale and magnitude as other simpler methods. Below is an annual time series of the additional flux amount calculated by the area-weighted method used in the Global Carbon Budget (a) and a similar plot showing the annual additional flux using the SeaFlux methodology (b). We have added this comparison figure to the revised manuscript in the Discussion section as we see the Global Carbon Budget interpolation as a primary potential use of a product such as SeaFlux.*

*For the area-weighting method, the interannual variability in this additional flux is a direct result of the IAV of the total global flux. Also, products with larger fluxes will have a larger correction inherently with this method, even if they aren't missing the largest area. For example, if two products were missing the exact same regions/gridcells, but one product had flux that was 0.3 PgC/yr larger for that year, the correction applied to the two products would be different, even though they were missing the same area. This assumes that the interannual variability of pCO2 in missing areas would be the same as for the rest of the product.*
*Another consideration is that simple area-weighting does not take sea ice cover into account, which is important given that the high latitudes are often the region lacking coverage. By first filling the product pCO2 maps with full spatial coverage and then calculating the flux, you account for this ice fraction.*

[Figure]

(a)                                                                          (b)

**R1: Figure 1A is not clear. Is "the changing fraction of area covered by observations" essentially seasonal changes in ice coverage? If so, perhaps include the different expressions for gas exchange in partial ice overage (e.g. Takahashi 2009), and different ice products in the analysis.**

*Response: We thank the reviewer for this comment as we had edited Figure 1 and neglected to update the figure caption sufficiently. These maps show the fraction of total months with coverage for each gridcell. Blue areas with a fraction of 1 represent regions that have coverage for all months of the product (here, 1988-2018). Yellow areas show where the product has no coverage for any months of the time series. We have amended the Figure 1 caption as such: Figure 1: Maps showing the fraction of months (1988-2018) with coverage available for each of the six pCO2 data products used in this study. Blue regions represent full temporal coverage of pCO2 in the product while yellow areas show regions with no reported pCO2 values for any month of the time series.*

**Minor issues:**
**R1: The gas transfer velocity is listed as piston velocity and exchange coefficient: be consistent**

*Response: Thank you for this comment. We have revised the manuscript and maintained consistency using the terminology "gas transfer velocity" and "coefficient of gas transfer" and removed piston velocity.*

**R1: I don't think that "improved" in the title is appropriate. The title for the SeaFlux product seem better as title for this paper: "SeaFlux data set: Air-sea CO2 fluxes for surface pCO2 data products using a standardised approach"**

*Response: Thank you for this suggestion. We have removed the word "improved" from the title and revised the title to: "SeaFlux: harmonization of air-sea CO2 fluxes from surface pCO2 data products using a standardised approach"*

**R1: Tables and figures are good but lines are difficult to read (for those with color impaired eyesight)**

*Response: Thank you for this comment. We have altered the colors on the figures to improve readability by those with color impairment by using the "colorblind" color scheme available from the Seaborn python visualization library (as shown in the figure below).*

[Figure]

---

## Author Comment (AC2)

**Review of "Harmonization of global surface ocean pCO2 mapped products and their flux calculations; an improved estimate of the ocean carbon sink" by Tim DeVries**

*We would like to thank Tim DeVries for the thoughtful comments and suggestions. In the following we will respond (in italics) to each comment.*

**The authors use a consistent set of input data to estimate the air-sea flux of CO2 from six different seawater pCO2 interpolation products for the period 1988-2018. They then fill in the missing areas in these data products using a scaled estimate of the air-sea CO2 flux from a recent data-based interpolation. The authors make two claims about the results: (i) that this methodology provides a consistent method for computing global air-sea CO2 fluxes from seawater pCO2 products, and that using this consistent methodology allows for improved intercomparison between the air-sea fluxes computed by different seawater pCO2 products, and (ii) that this methodology leads to improved estimates of the global air-sea CO2 flux as well as reducing the uncertainty in the global air-sea CO2 flux.**

**The first claim is certainly correct, and it will be very useful to have a common set of input parameters that modelers can use to determine the air-sea CO2 flux from their seawater pCO2 products. This will allow better intercomparability of results across different seawater pCO2 products. The input data and methodology are very clearly described in the paper, and seawater pCO2 modelers will find this to be a useful reference. On the strength of this aspect of the study, this is a useful study and should be published.**

**However, the second claim is misleading and requires substantial revisions to the paper. By using a uniform set of input parameters and assumptions, as well as a single product to fill in spatial gaps, the authors reduce the ensemble spread among the six different pCO2 products. However, this is not at all the same as reducing the actual uncertainty. The true uncertainty should reflect the uncertainty in the input parameters and assumptions used to compute the air-sea CO2 flux from seawater pCO2 products, and should also reflect the uncertainty of the air-sea CO2 flux in regions that are not covered by some products. Therefore, rather than stating that their methodology provides an "improved estimate" of the global air-sea CO2 flux with a reduced uncertainty, the authors should remove any such statements and instead explicitly discuss how their methodology artificially reduces the uncertainty in the global air-sea CO2 flux. This needs to be explicitly caveated, otherwise the community will cling to the numbers reported here as a "best estimate" of the global air-sea CO2 flux, which is not what it should be intended as.**
**I understand the desire to create a consensus estimate of the global air-sea CO2 flux, but this consensus will emerge when multiple independent methods yield the same answer, not when one approach is uniformly applied.**

*Response: Many thanks for the overall positive evaluation of our study and the helpful comments of caution and areas of improvement. Firstly, we have removed all mention of the word "improved" from our manuscript; we agree with the reviewer in that it is not so much the final mean flux estimate that is improved but the consistent methodology provided to the community that is the highlight of this work. We strive to better define the SeaFlux product as a full-coverage pCO2 product rather than an ensemble of flux estimates. The SeaFlux dataset is a resource to the community for calculating fluxes in a consistent manner is an additional aspect of this work and thus we do report calculated fluxes from the product but that is not the main result presented here.*

*We now respond to the reviewer's comment about reducing the ensemble spread among the products. While this methodology does slightly reduce the spread in the 6 products (reducing the global 1988-2018 mean flux spread from 0.54 PgC/yr based on product reported fluxes to 0.47 PgC/yr using the SeaFlux method) that is not the intention of the work. Instead we aim to provide a method to consistently compare the available products in a fair manner, specifically in a way that covers consistent areas and correctly applies the bulk gas flux equation. While the reviewer's statement that the SeaFlux method resulting in a smaller spread is not the same as reducing the uncertainty in the flux estimates, it is also true that the differences in flux resulting from inconsistent area coverage of the products does not represent "uncertainty" but rather is a result of comparing a complete global coverage map to an incomplete coverage map. One method to bypass this could be to cut down all the products only to the areas of the globe covered by all products. However, then the products wouldn't be comparable with global ocean models as their coverages would be significantly limited.*

*In response to the reviewers comments about uncertainty, throughout our revisions we take steps to account for uncertainty at each step of the flux calculation and report it explicitly including the inherent uncertainty in the observations themselves and the bulk flux parameterization equation. We have added Section 3.3 to the manuscript which specifically defines and addresses uncertainty intrinsic to the flux calculations themselves, the uncertainty introduced by our filling method, and the resulting spread of the products in the ensemble.*

*What we aim to provide with the SeaFlux dataset is a way to calculate $CO_2$ flux accurately and quickly and to make model-product global air-sea flux intercomparisons easier for the community. We also include each of the components for this flux calculation to maintain transparency for the user.*

*Lastly, we have removed the global mean flux estimate from the abstract, as Reviewer 1 pointed out that this is not in accordance with the scope of an ESSD article. We do include the value in the text but do not set it forth as a "best estimate" but just as the current flux estimate from this ensemble of products and the SeaFlux method.*

**R2: In addition to this concern about how the results are presented, I have a general concern with how the missing areas are filled in each product. Filling in the missing**

areas with the estimates of one single product (the MPI-ULB-SOMFFN) is problematic, as this implicitly assumes zero uncertainty in the gap-filled areas. Rather, these areas are precisely where the uncertainty is the largest, and indeed the method used here depends on several assumptions which have their own uncertainties (which of course is not reported in the uncertainty because it does not contribute to the ensemble spread). The best way to fill in these missing regions is to extend the individual methods to global coverage, as this would provide a better estimate of uncertainty in these regions. In general, I am concerned that using the MPI-UMB-SOMFFN product to fill in these gaps will become entrenched, and reduce the motivation for extending the various methods to global coverage. The authors should explicitly warn against this, and discuss the necessity of having multiple independent estimates of the air-sea CO2 flux in these regions (high latitudes, shelves/seas) in order to arrive at a good estimate of the uncertainty and an improved estimate of the global air-sea CO2 flux. In the meantime, the authors might want to consider using at least one other estimate of the air-sea CO2 flux in these missing regions — the Jena-MLS has global coverage and so could be used there as well.

*Response: We wholeheartedly agree with the reviewer that the "best way to fill these missing regions is to extend the individual methods to global coverage" and we are aware of at least 4 of the products working towards that goal currently. So we are highly confident that this method will not be "entrenched" or "reduce motivation". We have updated the text with mention of updated products having greater extent and how pySeaFlux could be adapted as that happens.*

*However, without currently published products with full coverage, we aim to provide a mechanism for the direct intercomparison between global means from models and products. And to do this the options are to either scale down all of the models and products to only where common areas exist, or fill in missing areas to get to common coverage, which is what we propose here.*

*We thank the reviewer for their suggestion of using the Jena-MLS product as an independent estimate of the missing regions. While this does seem like a plausible option from the full-coverage map shown, the Jena-MLS product is stated to be an "open-ocean" product. The creator of the Jena-MLS product (Christian Rodenbeck) specifically cautioned us against using the product as a coastal estimate, specifically stating that the Jena-MLS product is an open-ocean product.*

*For completeness, we calculated the resulting flux with the Jena-MLS product pCO2 used to fill in missing gridcells in the remaining products and present the resulting global mean CO2 flux time series here (dashed lines) along with the flux with filling by the SeaFlux package area-filling method employing MPI-UMB-SOMFFN (solid lines).*

[Figure]

**Specific Comments**

**R2: Line 24-25: It is true that the details of how these calculations are done varies greatly among methods, but it shouldn't be said that this unnecessarily enhances the uncertainties. Rather, the differences among methods reflects (only partly) the actual, true, large uncertainty that exists when converting sparse measurements of seawater pCO2 to global estimates of air-sea CO2 flux.**

*Response: We have revised this portion of the abstract by changing the word "uncertainties" to "spread". Within the text we discuss the uncertainty inherent to various methods of extrapolating sparse measurements of pCO2 to global coverage (the spread represented in the SeaFlux product ensemble) and distinguish it from the intrinsic uncertainty from the flux calculation itself (choice of wind product, wind speed parameterization, scaling of coefficient of gas transfer) which is now discussed in Section 3.3 of the manuscript.*

**R2: Line 25-26: Applying a uniform approach to all the different methods yields a lower ensemble spread, but it does not reduce the actual uncertainty. Filling in the gaps in products with a single product of course yields a lower spread (because a single estimate is applied to all the missing areas), which artificially lowers the "uncertainty". Likewise, applying a uniform gas transfer velocity and gas exchange formulation to all products artificially lowers the true "uncertainty".**

*Response: Throughout the manuscript we have been careful in our use of the term "uncertainty" and elaborated on the uncertainty estimate at each step in the flux calculation. We thank the*

*reviewer for these reminders and emphasize that through the harmonization approach of the SeaFlux ensemble we allow for improved intercomparison between products and models and provide a resource for the community to easily calculate ocean carbon fluxes from pCO2 mapping products.*

*To specifically address the suggestion that the filling method "artificially lowers the uncertainty", we calculate the mean product flux for the most conservative mask, only including gridcells that are included in every month for all six products. The spread of the products (as estimated by the standard deviation) for the 1988-2018 mean flux for this conservative mask is 0.173 PgC/yr). This is nearly identical to the spread of the final version SeaFlux ensemble product (std = 0.174 PgC/yr). Through this comparison, we would argue that the filling method does not in fact artificially lower the uncertainty or decrease the spread of the products.*

**R2: Line 31: Again, these "methodological inconsistencies" reflect our imperfect state of knowledge. Different groups chose different wind products, gas exchange parameters etc. Sure, one can repeat the calculations with a uniform set of parameters and get a lower spread, but one cannot argue then that this reflects a lowering of the true uncertainty.**

*Response: We have rephrased this portion of the abstract to incorporate the concerns of the reviewer. Here we were referencing the idea that using incomplete coverage maps to estimate a "global" value is inappropriate as is the use of a coefficient of gas transfer not scaled to the wind product utilized in the calculation. While our state of knowledge is indeed imperfect, there are many things that we have learned and the community should be implementing them appropriately in their flux calculations. A few specific things include the fact that NCEP1 and NCEP2 wind reanalysis products used with an unscaled coefficient of gas transfer can yield anomalous fluxes, much more so than the other wind products.*

**R2: Line 34: Regarding "appropriately scaled". This scaling factor has significant uncertainty. Also, it depends on the assumed relationship between wind speed and gas transfer velocity (i.e. linear, quadratic, cubic, or some mixture). Again, one can use a single number but it does not make the uncertainties go away.**

*Response: We agree. The scaling factor has an estimated 20% uncertainty (Wanninkhof et al 2014) and we have included discussion of this in the text. We have added the sentence "Wanninkhof (2014) estimates the error analysis for global carbon exchange to be 10% uncertainty in the coefficient itself based on the complex processing controlling the gas transfer velocity that is ultimately lumped into a single coefficient. The uncertainty estimate doubles to 20% when also including other cumulative factors which we refer to here as intrinsic uncertainty."*

*We also address the various wind speed parameterization relationships and note that we aim to expand the pySeaFlux package with additional parameterization options.*

**R2: Line 56-57: the "lack of systematic approach" is due to real uncertainties in how to do this calculation, not because some folks are doing it wrong**

*Response: We do not disagree that there are real uncertainties in the method of the calculation itself, however we would argue that using an unscaled coefficient of gas transfer or a coefficient that is scaled to a different wind product than the one used in a flux calculation is indeed incorrect. The uncertainty in the calculation is discussed and incorporated (Section 3.3) but using an incorrect calculation method does not add to "real uncertainties" but instead introduces needless error.*

**R2: Line 57-58: These differences don't "introduce uncertainty", they capture real uncertainty**

*Response: We direct the reader to the response above as it addresses this topic. We have revised this text in the manuscript to address the reviewer's concern.*

**R2: Line 60: Please clarify what you mean by "meaningfully compared". They can be compared regardless of whether or not these adjustments are made, one just has to attribute the differences correctly.**

*Response: We have edited this sentence in the text to more clearly explain our point which is the improvement of the consistency across the different products with respect to known and quantifiable shortcomings. Specifically they can be more equally compared with models and other products in a GBC or IPCC style approach.*

**R2: Line 79-80: Yes this is a consistent approach, which should be highlighted is useful for intercomparisons AMONG the different seawater pCO2 products, so that the differences between products can be attributed to differences in their underlying seawater pCO2 estimates. However, I would argue that a less consistent approach (i.e. one that better accounts for uncertainties) would be more appropriate for intercomparisons ACROSS products, e.g. for comparison with the biogeochemical models used in the Global Carbon Budget. If the SeaFlux products is used for cross-model intercomparison it will underestimate the true uncertainty.**

*Response: We acknowledge the reviewers point: the methodological inconsistencies, that are not necessarily incorrect choices, might better capture the uncertainties of an ensemble intrinsic to the flux calculation through the randomness of these choices. We address this in a discussion (in section 3.3), as this is a matter to which there is no specific solution. Key is that we make the reader aware of the implications of making comparisons across products.*
*The discussion will expand upon the true or intrinsic uncertainty in calculating fluxes that exists simply due to our incomplete understanding of the earth system. This includes the wind parameterization and scaling of the gas transfer velocity. There is also uncertainty added in the coastal filling method described in Section 2.1 and we discuss the magnitude of that within that*

*section. As mentioned in other comments from the reviewer, the filling method is not a perfect skill and our knowledge of coastal mechanisms is constantly improving.*

**R2: Line 97: It should be noted that there's no "problem" with any of the choices in Table A1, they are all reasonable assumptions. Notice they also all use the quadratic wind speed dependence, which is not necessarily "correct" (at least for all wind speeds)**

*Response: We have added a sentence to the paragraph elaborating on this: "While the choices made by each products creator, listed in Table A1, are not inherently incorrect, by utilizing a uniform methodology in flux calculation as provided by the pySeaFlux package, the differences in the resulting flux can be attributed to the $pCO_2$ mapping method itself.*

**R2: Line 116: It is important to say more precisely what you mean by "coastal". I think here you mean the continental shelf waters. So does not include the littoral zone, estuaries, tidal wetlands, seagrass/mangroves/kelp forests etc.**

*Response: We have added "and continental shelf waters" to this section of the text. We adopt the same definition for coastal as used by Landschutzer et al. 2020 (originally from Laruelle et al 2017 and from SOCAT, i.e. Bakker et al 2016) where coastal is defined following the broad SOCAT boundary definition of regions spanning from the coastline to 400 km distance from shore.*

**R2: General point about resolution: I don't think any of the products at 1 degree resolution capture "coastal" areas. So they are not missing coastal areas because of a lack of input data, but simply due to their coarse spatial resolution.**

*Response: We thank the reviewer for pointing this out and acknowledge that it could be the land mask used by the product creators that impacts the coverage near coasts and specifically that 1° gridcells (and even 1/4° resolution) could be missing key mechanisms in coastal regions. However, as seen in Figure 1, it is clearly not just the coastal region missing from many of these products and that's where the area-filling method is most impactful.*

*Additionally, we want to note that this resolution point also connects to another reason why the Jena-MLS product is not appropriate for filling missing products if our aim is to capture coastal processes. The original resolution mask of the Jena-MLS product is 4°x5° and therefore it is not fine enough to capture such processes.*

**R2: Another point: If the Jena-MLS does not require any filling, why not just fill the other products with the Jena-MLS product, instead of filling them with the MPI-ULB-SOMFFN product? Or at least, one could use both as an estimate of uncertainty.**

*Response: As stated in a previous response, the Jena-MLS product is stated to be an "open-ocean" product. The creator of the Jena-MLS product (Christian Rodenbeck) cautioned us against using the product as a coastal estimate, specifically stating that CarboScope (aka*

*Jena-MLS product) is an open-ocean product. This also strengthens our choice to use the MPI-ULB-SOMFFN product to fill the missing continental shelf waters, since Landschützer et al. (2020) explicitly predict pCO2 for these waters at a 0.25° scale. Further, the predictor variables that the authors use are tailored to estimating coastal pCO2 in the coastal waters, e.g., depth is used as a predictor (see e.g. Laruelle et al 2017), which none of the open ocean only methods use.*

**R2: Line 129-130: This is confusing. The way that the regions are defined shouldn't matter in terms of calculating a global air-sea CO2 flux. Do you mean differences in the coverage or lack thereof in different regions?**

*Response: We have removed this sentence from the paragraph. It was referencing a finding using different shelf definitions from Shutler et al 2016, but with the revised manuscript it no longer fit into the discussion.*

**R2: Line 135: What is the spatial resolution of the MPI-ULB-SOMFFN product?**

*Response: The spatial resolution of the merged climatology product is 1° x 1° in open ocean regions and 0.25° x 0.25° in coastal regions. Again, coastal is defined as coastline to 400km distance from shore. In order to seamlessly merge the datasets at the boundary between the two, Landschutzer et al. (2020) use a 3 step approach described in Section 2.2 of that manuscript. The final merged climatology is at 0.25 degree spatial scale and we upscale to 1x1 degree resolution for use with the products.*

**R2: Table 1: What resolution grid is used to calculate the area? Please also define the land/sea mask**

*Response: The area is calculated at a 1° x 1° resolution, since this is the same as the data products. The code used to calculate the area is in the pySeaFlux package. The "mask" is based on the coverage of the MPI-ULB SOMFFN product. We mask inland and freshwater bodies out. During revisions we have updated the product to utilize the ETOPO1, available at https://doi.pangaea.de/10.1594/PANGAEA.769615 (Amante and Eakins, 2009).*

**R2: Table 1: It would be good to report the actual global mean pCO2 before and after the filling is applied**

*Response: Thank you for this suggestion. Currently, in Table 1, column 2, we show the adjustment in global mean pCO2 from the filling for each of the six products. We have added in parentheses below that number, the original global mean pCO2 and the filled global mean pCO2 as suggested by the reviewer.*

| Product | Area coverage | Mean Global $pCO_2$ change (µatm) | Northern Hem $pCO_2$ change (µatm) | Southern Hem $pCO_2$ change (µatm) |
|---------|---------------|-----------------------------------|------------------------------------|------------------------------------|

|  | (% global ocean) |  |  |  |
| --- | --- | --- | --- | --- |
| CMEMS-FFNN
*Denvil-Sommer et al. 2019*
*Chau et al. 2020* | 89% | -1.68
(362.25/363.93) | -4.35
(359.93/364.28) | 0.30
(363.98/363.68) |
| CSIR-ML6
*Gregor et al. 2019* | 93% | -0.923
(361.78/362.70 | -2.15
(359.40/361.55) | 0.07
(363.54/363.47) |
| JENA-MLS
*Rödenbeck et al. 2013* | 100% | 0.00
(359.97/359.97) | 0.00
(355.77/355.77) | 0.00
(363.08/363.08) |
| JMA-MLR
*Iida et al. 2020* | 85% | -0.69
(359.72/360.41) | -2.443
(357.14/359.58) | 0.77
(361.63/360.86) |
| MPI-SOMFFN
*Landschützer et al. 2014*
*Landschützer et al. 2020a* | 89% | -1.07
(362.02/363.09) | -2.62
(359.71/362.33) | 0.16
(363.72/363.56) |
| NIES-FNN
*Zeng et al. 2014* | 92% | -0.36
(360.28/360.64) | -1.965
(359.75/361.71) | 0.90
(360.67/359.77) |

**R2: Equation (2): The authors should plot this scaling factor out over time for the different models.**

*Response: Thank you for this comment. There is in fact only one scaling factor which is applied to the climatology. Then this scaled climatology is used to fill in the missing areas of each product. So if multiple products are missing pCO2 values for the area 80-90N; 0-10E, for all of 2016 for example, all of those models will be filled with the same values for that area and time period. The filling does not differ by product. We have elaborated on this further in Section 2.1 to ensure this method is clear.*
*Figure 2 has an inlay of time series of the scaling factor \* mean pCO2 from the climatology. We have also included a time series of the scaling factor itself here for your reference.*

[Figure]

**R2: Paragraph starting line 155: This application of the scaling factor could be problematic for several reasons. First, the scaling factor is derived for the open ocean, but is applied in the coastal ocean (i.e. shelves and seas). As the authors mentioned, the processes driving the pCO2 in the shelf/sea regions are distinct from those in the open ocean, so this scaling factor might not be appropriate. It would be better to use a scaling based on results from a high-resolution carbon cycle model that includes shelf/sea regions.**

*Response: We thank the reviewer for this idea. We do acknowledge (paragraph starting Line 165) that this method is not without its own assumptions, one of which is pointed out here by the reviewer- that the scaling factor method assumes that the missing areas are adjusting temporally in the same way as the open ocean. However, we reason that it would be better to continue to use the MPI-ULB-SOMFFN approach for filling since it is methodologically consistent with the other gap-filling methods. Using a high-resolution model that includes the coastal ocean would require an independent study to decide on the model that is able to accurately represent the carbon cycle in the coastal ocean and that is outside of the scope of this study.*

**R2: Second, this implicitly assumes no interannual variability in the shelf/sea carbon sink (leading to too low uncertainties). This is problematic because the shelf/sea regions are highly impacted by human activity such as fishing/trawling/farming etc. So I think there are very large unreported uncertainties that arise due to this scaling factor.**

*Response: As shown in the figure above representing the time series of the scaling factor, it is clear that there is interannual variability in this method. Again, it is just assumed that the interannual variability of the shelf/coastal region is the same as that of the open ocean. While we agree that the shelf and coastal areas are indeed impacted by human activities as mentioned here, the impact of those variations would be washed out by the dominating signal of the open ocean variability when looking at these global mean values. If someone were especially interested in coastal trends, this method would not be suitable to extrapolate the sparse observation coverage to longer time series.*

**R2: As asked before: Why not just fill the other products with the Jena-MLS product? Then this scaling would not be needed.**

*Response: We have addressed this question above, but to restate, the Jena-MLS product is inherently an "open ocean" product and the creator specifically cautions against using the product as a coastal estimate despite it's full global coverage.*

**R2: Lines 166-168: Not true, the Jena-MLS product covers those regions.**

*Response: We have edited this paragraph to include discussion of the Jena product and why it is not suitable to use for coastal/continental shelf region filling.*

**R2: Line 175: The area-based scaling could be called a "reasonable first order approximation" as well.**

*Response: We thank the reviewer for this comment. Indeed, as shared elsewhere, the area-based scaling is on the same order of magnitude as this method, but the SeaFlux method provides the opportunity for key improvements over that method. For the area-weighting method, the interannual variability in the additional flux is a direct result of the IAV of the total global flux. Also, products with larger fluxes will have a larger correction inherently with this method, even if they aren't missing the largest area. For example, if two products were missing the exact same regions/gridcells, but one product had flux that was 0.3 PgC/yr larger for that year, the correction applied to the two products would be different, even though they were missing the same area. This assumes that the missing area would be anomalous in the same way that the rest of the product is. Another consideration is that simple area-weighting does not take sea ice cover into account, which is important given that the high latitudes are often the region lacking coverage. By first filling the product pCO2 maps with full spatial coverage and then calculating the flux, you account for this ice fraction.*

*We have extended this paragraph in the manuscript to discuss the added benefits provided by this approach.*

**R2: Line 197: Please clarify what is meant by "second moment of the average"? The average is the first moment.**

*Response: This has been changed to the "the square of the wind speed".*

**R2: Line 198ff: The quadratic dependence is most often used, but the actual relationship could vary from less than linear (Krakauer et al., 2006) to almost certainly cubic at high wind speeds due to bubble-mediated transfers (Stanley et al., 2009), so this is a significant source of uncertainty.**

*Response: We have added discussion of alternative wind speed parameterizations to Section 2.3 of the manuscript (as quote above in a previous response). Additionally, we are in the process of adding alternative options for this calculation to the pySeaFlux package, specifically a cubic parameterization. While this choice definitely adds additional uncertainty to the final flux value, it is common practice, in both models and product analysis, to choose one parameterization for flux calculations. We discuss the impact alternative choices would have on the resulting flux and therefore the added uncertainty, but one choice must be made for the ultimate calculation.*

**R2: Line 212: "piston velocity" is used here, while elsewhere it is "gas transfer velocity"**

*Response: We thank the reviewer for pointing this out. We have edited the entire manuscript to be consistent with such terminology. We are now using the terminology "gas transfer velocity".*

**R2: Line 217-218: This was not clear to me. What is meant by "a probability distribution of wind speeds is used to optimize the gas transfer coefficient"? Also, it should not be stated that the rate of bomb 14C invasion is observed — rather, it is inferred from an estimate of the bomb 14C inventory in the ocean (which has a significant uncertainty), and also requires the intermediary of an ocean circulation model (another source of uncertainty).**

*Response: The section on the scaling of kw based on the bomb 14C inventory will be revised. As the reviewer points out, there are several sentences that are unclear and need to be rewritten. This applies to lines 217 through 232.*

**R2: Lines 219-220. This sentence is also unclear. How would you scale a gas transfer coefficient to a bomb-14C inventory? Maybe what the authors mean is that the estimated bomb 14C inventory has been used to infer a global average estimate of the gas transfer velocity, and that different methods have used gas transfer velocities that may not be**

**exactly the same as ones that have been inferred in previous studies (e.g. Sweeney et al or Naegler et al).**

*Response: Thank you for this comment and suggestion for clarifying this sentence. We have edited the section to read, "Further, while estimated bomb-14C inventory has been used to infer a global average estimate of the gas transfer velocity, different products have used gas transfer velocities that may not be exactly the same as ones that have been inferred in previous studies (e.g. Sweeney et al 2007; Naegler et al. 2009) (Table A1). We have also added a statement in Section 2.3 discussing the estimated 20% uncertainty on this method from Wanninkhof 2014. We also touch on this idea of cascading uncertainty from each of these choices in the added Section 3.3 which specifically discusses uncertainty in the SeaFlux ensemble product. By choosing to scale each product to a set kw value (16.5 cm/hr here) we are potentially artificially reducing the uncertainty in the flux through this step. However, each of the products does this scaling independently in their original flux releases (Table A1) and nearly all of them opt to use the same scaling methodology.*

**R2: Lines 220-221: This is not clear again. What do the authors mean that "the range of bomb-14C estimates is within the range of uncertainty from the associated studies"? It sounds like the authors mean that they used the gas transfer coefficients and wind speed parameterizations from the various studies listed in Table A1 and then calculated the bomb-14C inventory in the ocean, and compared that to the estimated value of the bomb 14C inventory from Naegler (2009). But I doubt that is the case because the bomb spike goes back to 1955 and it requires a model for the seawater bomb 14C as well (i.e. a circulation model). So maybe what they mean is that the globally-averaged gas transfer velocity used by the different studies listed in Table A1 is within the range of uncertainty on that particular parameter that has been deduced from studies that have inferred the globally averaged gas transfer velocity using an estimate of the bomb radiocarbon inventory and a circulation model (e.g. the estimates of Naegler, 2009).**

*Response: We have rephrased this using the reviewer's suggestion.*
***Before:*** *The range of the different bomb-14C estimates is within the range of the uncertainty from the associated studies (Naegler, 2009), but the choice would introduce inconsistency that is easily addressed here.*
***After:*** *The range of globally-averaged kw that previous studies scaled to (Table A1) is within the range of uncertainty for globally-averaged kw estimated from bomb-14C based estimates as reported by Naegler (2009). We choose a single value to scale kw to (16.5cm/hr) which ultimately reduces the spread of flux estimates, but not that this does not reduce the uncertainty which Naegler (2009) reports at roughly 20%.*

**R2: Equation (4): Here the "a" parameter is not the same one as used in equation (3), because you have introduced the (1-ice) term in this equation. If you say that Kw (equation 3) includes the (1-ice) in its definition, then equation (4) would be divided by (1-ice), and you would eliminate the (1-ice) from equation (1). So you need to say**

whether you are finding a value of "a" that yields a value of Kw as defined by equation (3) that is 16.5 cm/hr on average, or whether you are finding a value of "a" that yields a value of Kw*(1-ice) that is 16.5 cm/hr on average. Also, notice the additional uncertainty that this choice introduces.

*Response: Thank you for this comment. In Equation 4, we are estimating the coefficient of gas transfer using a set bomb-14C flux estimate. We have edited this calculation and eliminated the (1-ice) term. This is consistent with Wanninkhof 2014.*

**R2: Lines 231-232: What do you mean "even with the same bomb-14C observations the scaled coefficient (a) can have a 40% range?" This sentence is puzzling because the coefficient "a" is completely independent of the bomb-14C observations. Do you mean that the value of "a" that would be inferred by an inverse model that tried to match the bomb-14C inventory in the ocean would depend on the wind speed product used by that model, such that using different wind speed products could result in optimal values of "a" that are as much as 40% different from one another?**

*Response: Thank you for this question. What we mean is that even when using the same global mean gas transfer rate to scale the coefficient in Equation 4 (here 16.5 cm/hr), the uncertainty estimate of k for global or basin-scale applications is 20% (Wanninkhof 2014). And further, global average winds from various wind speed reanalysis products have a considerable range (Naegler et al. 2006) and can yield a 40% range in the coefficient, even with the same 14C constraints. We have edited the sentence to read: "Global mean winds from the various wind speed reanalysis products have a considerable range and therefore even when utilizing the same bomb-14C constraints, the scaled coefficient (a) can have a 40% range (Wanninkhof 2014)."*

**R2: Line 232-233: No, one cannot reduce the uncertainty in the global fluxes in this way (unless the original value used falls outside the uncertainty bounds of the average gas transfer velocity). One can reduce the ensemble spread by specifying that each model use the same globally-averaged gas transfer velocity. This is not the same as reducing the uncertainty. The true uncertainty would take into account the uncertainty in the value of "a" (which is ~20%, see Wanninkhof, 2014) as well as the uncertainty in the form of the gas transfer velocity parameterization itself (e.g. quadratic vs. cubic).**

*Response: Thank you for this comment. We have clarified throughout the manuscript that it is the apparent uncertainty that we are reducing with the resources provided in the pySeaFlux package and through the resulting SeaFlux data product, not the inherent uncertainty intrinsic to the equations themselves. We have revised the manuscript and added uncertainty discussion and quantification in each subsection to help separate the apparent uncertainty versus the intrinsic uncertainty.*

**R2: Line 240: The authors say "our results show" and then cite the study or Roobaert et al. (2018). Please clarify which study shows this, the present one or that of Roobaert et al.**

*Response: It is indeed results from this study that result in the stated 9% spread in resulting fluxes. We have removed the reference to Roobaert et al. (2018) but there remains a reference to that study in the Introduction where we discuss their presentation of uncertainty in air-sea carbon flux induced by various parameterizations of the gas transfer velocity and wind speed data products.*

**R2: Line 248-249: What do you mean by "small, but not insignificant"? Can you state the magnitude of the impact?**

*Response:  The magnitude of the impact that the pH2O correction has on the fluxes will be quantified and the value will be stated.*

**R2: Line 261: Taking a global mean is straightforward regardless if you account for spatial coverage differences or not. So I think the authors mean something other than "straightforward".**

*Response: I disagree with this statement. While the mathematical equation of calculating a mean is indeed straightforward, when consider fluxes, the total global area must also be considered. Often a set global mean area will be used in the calculation, so that it is consistent between all products and models for example. However, if the product itself doesnt cover the full global area, then it is a misleading and incorrect global mean. So by filling the products to cover a common seamask, the global mean calculation becomes straightforward. We appreciate the reviewers comment though and have rephrased this sentence to remove the word "straightforward".*

**R2: Line 265-267: Please state the spread due to wind product without scaling, vs. with scaling**

*Response: We include the difference in mean flux for the product ensemble in Table 2, for each of the wind products. We have included a reference to Table 2 in this sentence for clarity.*

**R2: Line 269-270: Yes, the SeaFlux allows "a more accurate comparison of fluxes" within pCO2 products, but it does NOT lead to "increased confidence"**

*Response: Thank you for this comment. We have removed the term "and increased confidence" and have edited the manuscript to highlight the method itself rather than an specific ability for it to increase confidence in the global mean flux estimate itself.*

**R2: Figure 5: It would be useful to show the individual models before and after the corrections are applied. Why does the NIES-FFN appear to change so much relative to the others in the later period? Is it due to the gas exchange scaling or infilling?**

*Response: Below is a figure with subplots for each product. The product-creators reported flux is in dashed while the SeaFlux product flux values are in solid lines.*

[Figure]

**R2: Lines 275-276: Actually, one should have lower confidence in the uncertainties and the ensemble mean, because the uncertainties have been understated. Also, the phrase "higher confidence in the uncertainties" sounds a bit strange because it implies you are reducing the uncertainty in the uncertainty.**

*Response: We agree that this sentence is inaccurate and have removed it. We have addressed the sentiment of the reviewer's concern by being more explicit in the fact that we are not reducing the uncertainties - specifically in this case where we are only scaling kw to only 16.5 cm/hr while there is still a 20% uncertainty to this global estimate.*

**R2: Line 278-280: "working towards consensus on other issues" implies that there is a**

**consensus on the issues addressed here (gas exchange parameterization, shelf/sea CO2 flux). However, there is not a consensus on this, the authors have simply picked one reasonable approach to these issues and applied it uniformly to different seawater pCO2 products. A consensus will emerge when multiple independent methods yield the same answer, not when one approach is uniformly applied.**

*Response: We thank the reviewer for this comment and agree that a consensus has not been reached with respect to gas exchange parameterization etc. We have reworded this sentence to more accurately reflect the topics addressed (skin surface correction and river efflux estimation). We have edited this sentence to read, "While the pySeaFlux package presents one approach to standardize much of the calculation of air-sea carbon flux, there remains many additional issues that the ocean carbon community is still working towards understanding and incorporating (i.e. skin temperature effect, river efflux)."*

**R2: Line 318: The authors should explicitly discuss/restate here how this uncertainty estimate is too low. It would be unfortunate if the community mistook this as a consensus estimate or "best estimate" of the mean and/or its uncertainty.**

*Response: We thank the reviewers for their suggestion. We have revised the entire manuscript and focus on the data product (that can be used to calculate $FCO_2$ from $pCO_2$) rather than the resulting "best estimate" of global carbon flux. We have removed the term "best estimate" from the manuscript. We also include uncertainty at each step and discuss how this apparent uncertainty estimate omits intrinsic uncertainty and therefore is itself an under estimate of the total uncertainty.*

**R2: Line 324: What do the authors mean by this "may reduce the current carbon budget imbalance"? This needs to be spelled out in more detail.**

*Response: This statement is in reference to the carbon budget imbalance reported in the Global Carbon Budget. We have revised this sentence to clarify that point: "...may help reduce the current carbon budget imbalance reported in the Global Carbon Budget..."*

**R2: Line 329-330: Since others will be able to apply these "standardizations" to their datasets, I think this should come with a warning. These standardizations should be used for applications where there is going to be intercomparison with other seawater pCO2 products. But it should not be used as a method to arrive at a "best estimate" of the air-sea CO2 flux. What the community really needs is independent methods of estimating Fnet in order to derive a robust estimate of the mean and uncertainty. If everyone uses the same methods, assumptions, and datasets, we will never know what the true answer is.**

*Response: Our aim is to lower the spread in the ensemble of pCO2 products that result from choices made in the flux calculation to isolate the uncertainty due to the mapping techniques themselves. Each of these products uses their own interpolation mechanism (neural network,*

*multiple linear regression, etc) and despite being based on the same dataset (SOCAT), they all come up with different pCO2 estimates. In that sense, they are representing independent methods of estimating the flux. By offering a method to standardize the flux calculation, and presenting an ensemble mean global flux estimate here, we highlight the uncertainty from the pCO2 products themselves, outside of the uncertainty intrinsic in the calculation of flux from available surface ocean pCO2 observations.*

*We do purport that these estimates can also be compared to an ensemble of global ocean hindcast models which also estimate flux. The same assumptions are made in the creation of these models with regard to choosing a wind speed parametrization for flux. We aim not to squelch future work to improve global flux estimates, but to provide the community a tool to calculate fluxes for intercomparison efforts.*

---

## Author Comment (AC3)

**Review of "Harmonization of global surface ocean pCO2 mapped products and their flux calculations; an improved estimate of the ocean carbon sink" by Andrea Fassbender**

**Fay and coauthors aim to improve the global net air-sea CO2 flux estimate and ease model-data comparisons by making a diversity of pCO2 data products (n=6) with methodological differences more consistent and releasing the results as a new data product: SeaFlux. Their approach involves relying on a climatological pCO2 data product to spatially extrapolate estimates from other pCO2 data products with more limited ocean coverage. After extrapolating all pCO2 data products to the same ocean mask, the authors calculate the net air-sea flux using three wind speed products, while accounting for gas exchange coefficient sensitivities to the individual wind speed products. The authors find that the flux estimate discrepancies between these products can be reduced most by simply using a consistent ocean domain for the pCO2 data products.**

**The paper is clearly written, the findings are important, and the data product will simplify model-data comparisons. However, a justification for the extrapolation approach is not provided, and a few simple analyses are required to verify that the approach is "a step forward from" existing methods. Reviewers 1 and 2 have already outlined several concerns; therefore, I will keep this brief and focus on specific suggestions and technical corrections for the authors to address.**

*We would like to thank Andrea Fassbender for the thoughtful comments and suggestions. In the following we will respond (in italics) to each of her specific comments.*

**Specific Comments**

**R3: My primary concern is with the pCO2 scaling approach. It is not clear why the MPI-ULB-SOMFFN climatology was used for gap filling rather than the time evolving JENA-MLS data product. As noted by Reviewer 2, the authors could test their scaling method by using JENA-MLS as the reference data product to see if they achieve similar results.**

*Response: Thank you for this question. We will respond, as we did with Reviewer 2, that we are aware of at least 4 of the products working towards the goal of full spatial coverage. But until those are released and available we have to work with what we have.*

*We thank the reviewer for their suggestion of using the Jena-MLS product as an independent estimate of the missing regions. While this does seem like a plausible option from the full-coverage map shown, the Jena-MLS product is stated to be an "open-ocean" product. The creator of the Jena-MLS product (Christian Rodenbeck) specifically cautioned us against using the product as a coastal estimate, specifically stating that CarboScope (aka Jena-MLS product) is an open-ocean product. Additionally, the Jena-MLS product is released in its native resolution as a 4x5 degree map and we have downscaled it to 1x1 degree for this ensemble work.*

*Therefore, it is not produced in a sufficiently fine-scale resolution to capture coastal and continental shelf processes. The MPI-ULB-SOMFFN climatology is produced at 0.25 degree spatial scale at the coastal margins (up to 400km off land) and therefore provides an improved representation of these regions.*

*For completeness, we did calculate the resulting flux with the Jena-MLS product pCO2 used to fill in missing gridcells in the remaining products and present the resulting global mean CO2 flux timeseries here (dashed lines) along with the flux with filling by the SeaFlux package area-filling method employing MPI-UMB-SOMFFN (solid lines).*

[Figure]

**R3: It is also unclear why the authors use an ensemble mean scaling factor when individual scaling factors for each data product may be more appropriate as it would allow more data to be used (i.e., a consistent mask wouldn't be required) to determine the scaling factor for most products. I can understand the desire to maintain consistency in the data extrapolation between products, but it's not clear that this approach makes more sense than creating individual scaling factors for the data extrapolations. More information is needed to explain why this decision was made.**

*Response: We considered the option of calculating individual scaling factors but opted to present the ensemble approach for it provides strengths with regard to capturing a forced signal common to all products. We acknowledge that a scaling factor for each product would be much more sensitive to interannual variability in the products however we don't see that as a strength when considering long-term global mean averages. We present here a comparison of a scaling factor for various products to that of the ensemble mean.*

[Figure]

*Scaling factors for the ensemble product (thick grey line). We only show NIES, Jena and JMA methods. The remaining methods are more similar to the ensemble scaling factor than the methods shown in the figure above. The JMA-MLR approach shows the largest difference from the ensemble mean, which is on average 0.6% (max of 1.7%) for pCO2. When propagated through to fluxes, this equates to an average difference of 0.04 Pg yr[1] when compared with the scaling factor computed with the ensemble. For reference, filling results in a ~0.25 Pg yr[1] larger uptake by the ocean (as shown in the figure on the following page). Thus, even in the worst case scenario, the difference is small.*

*Further, using the ensemble approach makes the SeaFlux data product less product specific, as new surface pCO2 products can also apply the approach without having to perform the calculation.*

**R3: As a sensitivity test, the authors could apply their methodology using JENA-MLS to scale MPI-ULB-SOMFFN (and vice versa) directly AND using an ensemble mean, to see which yields a better result. They could also do this using (1) the common missing data mask as well as (2) each missing data mask from the four other data products to evaluate whether the resulting extrapolation bias is sensitive to the extrapolation area. The authors could also apply the linear-scaling approach used in the Global Carbon Budget (GCP) to MPI-ULBSOMFFN and JENA-MLS (using the missing data masks from the four other products) to quantify the resulting extrapolation biases and determine whether their approach is indeed more accurate than the GCP method. The suggested analyses may help clarify which approach is best for achieving data product comparability.**

*Response: We thank the reviewer for these ideas.*

*For the first comment regarding a sensitivity test, we have shown the comparison in resulting flux when using the Jena-MLS product to fill in missing areas of all the other products, however*

*we remind the reviewer that this is simply a comparison but not the preferred method as Jena-MLS is an open-ocean product and should not be used as a parameterization for coastal regions. However we also question how one would define "a better result" as suggested by the reviewer as the "truth" is of course unknown.*

*To respond to the idea of applying the linear-scaling approach used in the GCB, one strength of the SeaFlux approach is that it deals directly with the pCO2 in the various pCO2 interpolation products rather than adjusting their fluxes after the fact. These products are created to represent (nearly) full coverage pCO2. They often will also calculate a flux field and report it as well, but the approach itself is developed and tuned to represent pCO2 values. So by calculating a global flux and then adjusting it by a missing area factor you are skipping over the point that there are pCO2 values missing and how that unknown could cascade in the flux calculation itself. Here we show that the additional flux for each product resulting from the area-filling method proposed here in the SeaFlux package is on the same scale and magnitude as the method utilized by the GCB. Below is an annual time series of the additional flux amount calculated by the area-weighted method used in the Global Carbon Budget (a) and a similar plot showing the annual additional flux using the SeaFlux methodology (b).*

[Figure]

(a)                                                                          (b)

**Technical Corrections**

**R3: Title: The 2 in pCO2 should be subscripted.**

*Response: Thank you for this correction.*

**R3: Line 36: pCO2 is not yet defined.**

*Response: Thank you for pointing this out. We define pCO2 in the first sentence of the Introduction but opt to not define it in the abstract as we aim to be as concise as possible in that section.*

**R3: Line 37: Add "modern" before "global mean uptake"**

*Response: Thank you for this suggestion. The referenced sentence has been removed from the manuscript but we have added this designation in other sections.*

**R3: Line 43: "variations" should be "variation." It seems that the atmospheric pCO2 growth
rate is the largest driving force governing the net exchange of CO2 across the air-sea interface unless you're talking about sub-annual or pre-industrial timescales. Please clarify.**

*Response: We have edited this sentence to include the importance of the growing atmospheric pCO2 levels and its variability as the driving force for variability in surface ocean pCO2 levels.*

**R3: Line 57: How about: "These differences in flux calculations introduce uncertainty in comparisons between the products and with Global Ocean Biogeochemistry Models (GOBM)."**

*Response: We have changed the sentence to read as suggested.*

**R3: Line 95: pCO2 was already defined.**

*Response: Thank you for pointing out this repetition. We have omitted the definition.*

**R3: Line 97: A "we" seems to be missing.**

*Response: We have added the missing word.*

**R3: Line 100: Satellite SST and EN4 subsurface salinity data are used to calculate parameters required for the air-sea flux calculations. What depth are the EN4 salinity data from?**

*Response: We have included that we use the near-surface salinity from EN4.2.1 that is an estimate of salinity at ~5 m.*

**R3: Line 108: Slightly awkward wording. What about: "Flux is defined as being positive when CO2 is released from the ocean to the atmosphere and negative when CO2 is absorbed by the ocean from the atmosphere."**

*Response: We thank the reviewer for this suggestion and have reworded the sentence as such.*

**R3: Line 117: "…relationships between pCO2 and proxy variables are expected". The next sentence starting on this line doesn't seem to make sense. Maybe get rid of "in contrast."**

*Response: Thank you for this suggestion. We have removed the "In contrast" from the beginning of that sentence and improved the flow between the two sentences.*

**R3: Line 130: "net global(?) fluxes"?**

*Response: This sentence has been removed from the manuscript.*

**R3: Line 159: There seems to be a formatting issue. Additionally, it's not clear if you are talking about the original global flux for each model, or not.**

*Response: Thank you for this comment. We have amended this sentence to make clear that we are discussing the impact of the area-filling method on the resulting global mean flux values for each of the six products, and the product ensemble mean.*

**R3: Line 160: Is this because some products are missing the Arctic? That seems important to clarify.**

*Response: Yes, the Arctic is a common area missing from most pCO2 product maps, but also there are a few products that are missing portions of the North Pacific region (Figure 1) which also adds to this adjustment. We have added sentences to further discuss this large adjustment in the Northern Hemisphere for specific products.*

**R3: Line 162: "…the final CO2 flux also depends on the…"**

*Response: inserted "also" in the sentence: "...the final CO2 flux **also** depends on the…"*

**R3: Line 182: Remove equal sign.**

*Response: Done.*

**R3: Line 330: Typo.**

*Response: Revised to "...presented standardized flux calculations to **their** own data-based pCO2 reconstructions."*

**R3: Figure 2: I would recommend converting this a four-panel figure with the inset graph having its own panel since there is space. Add a-d lettering to match the caption.**

*Response: We have converted this figure to a 4 panel figure as suggested by the reviewer.*

**R3: Table 1: It is not clear what "unfilled area listed" means. Should this be "Area coverage"?**

*Response: We have revised this sentence in the Table caption to read "Area coverage listed represents average annual area covered for 1988-2018 as this value changes monthly for many products."*

**R3: Table 2: Should the 3rd row be titled "Mean Annual Global Flux"**

*Response: We assume that the reviewer is referring to the 3rd column in this table, currently labeled "Mean flux difference: scaled – unscaled wind". It is indeed the difference in mean 1988-2018 flux when using a scaled coefficient of gas transfer versus a set coefficient of 0.26. We have added to the Table caption to clarify what is reported within: "Column 1 lists the scaled coefficient of gas transfer for each of 3 wind reanalysis products; column 2 includes the global mean flux using each wind product. Column 3 shows the difference in resulting flux when using a scaled coefficient of gas transfer versus a set value of 0.26."*

---

## Referee Report (RR1)

**General Comments:**

I thank Fay, Gregor, and coauthors for diligently addressing my prior concerns with edits and/or clear explanations. I have provided a few minor comments for the authors to take or leave. These minor suggestions touch on places in the manuscript where I find the text to be somewhat unclear (relative to the rest of the manuscript), which could be easily addressed with slight wording changes.

**Minor Suggestions:**

**Abstract:** I think the abstract could be streamlined. Line 23 says " this resource" but a resource has not yet been introduced, so it's not immediately clear what is being referenced. In the next sentence, a dataset is referenced, yet this also has not yet been introduced. Some minor rearrangement of existing abstract text to introduce the resource and dataset before describing their utility would go a long way toward clarifying the useful products/tools being presented.

**Line 38:** "*This is because **long term** variation in surface ocean pCO2, ultimately driven by increases in atmospheric pCO2 levels, is the driving force governing the exchange of CO2 across the air-sea interface,..*"

On shorter timescales, interannual and seasonal variations can be the dominant driving forces.

**Eqn. 1:** I noticed that the Zenodo data product description uses $K_0$ rather than sol.

**Line 57:** "*The resulting flux estimates can then be more directly compared with respect to uncertainty attribution with no source of difference that is not implicit in the mapping method or flux calculation.*"

I find this sentence unclear. Are you suggested that filling gaps with a scaled climatology makes it easier to compare/attribute uncertainties between products? I think you mean that gap filling makes is easier to isolate mapping differences rather than area related differences in fluxes, but it's not 100% clear. Also, the wording "no source of difference that is not implicit…" is difficult to follow (here and on and line 308). What about – all sources of discrepancy can be attributed to the mapping method…

**Line 128:** This is oddly worded and it's not clear if the pCO2 or flux data are scaled. What about:
To account for differing area coverage, past studies (Friedlingstein et al. 2019, 2020; Hauck et al. 2020) have adjusted observation-based flux products by simply scaling the datasets based on the percent of the total ocean area represented.

**Line 171:** "*Globally, the area-filling adjustments result in a difference of less than 17% of the total flux in all products, with the mean adjustment for the six products at 8%.*"

This wording is quite different from the framing in the abstract. The main text minimizes the differences while the abstract emphasizes them. Abstract: "*We address differences in spatial coverage of the surface ocean CO2 between the mapping products which ultimately yields an increase in CO2 uptake of up to 17% for some products.*"

**Line 280:** I'm not sure that I agree with this sentence – or I may be missing something here. Table 2 suggests that wind speed differences (column 3) cause a 0 to 0.15 difference in global mean flux while scaling (column 4) can cause a -0.04 to -0.13 difference in global mean flux (presumably also units of PgC yr-1). I guess if you use the average from these columns this sentence makes sense – but I wouldn't say

it's clear. Relatedly, if the scaled coefficient for JRA55 is 0.26, should we see differences between fluxes calculated with this scaled value and the fixed values of 0.26 when they are identical?

**Line 321:** "*Secondly, the cool skin correction would be equally applied to all methods and would not contribute to the inconsistencies in flux calculation that we are trying to address here*."

The influence of temperature on pCO2 is dependent on the starting pCO2 value, so different pCO2 values in the mapped products would result in different cool skin adjustments (even if the temp/sal adjustment were constant) – though this would likely result in very small differences between products in most regions. For accuracy, maybe rephrase to say that this effect would negligibly contribute to the inconsistencies in flux calculation being addressed?

**Line 326:** I think Fnet is used twice and Fant is used once. Maybe remove?